# Studying the Human Microbiota: Advances in Understanding the Fundamentals, Origin, and Evolution of Biological Timekeeping

**DOI:** 10.3390/ijms242216169

**Published:** 2023-11-10

**Authors:** Adam Siebieszuk, Monika Sejbuk, Anna Maria Witkowska

**Affiliations:** 1Department of Physiology, Faculty of Medicine, Medical University of Bialystok, Mickiewicza 2C, 15-222 Białystok, Poland; adam.siebieszuk@gmail.com; 2Department of Food Biotechnology, Faculty of Health Sciences, Medical University of Bialystok, Szpitalna 37, 15-295 Białystok, Poland; sejbuk.monika99@gmail.com

**Keywords:** biological clock, evolution, microbiome, metabolic oscillations, peroxiredoxins, cellular timekeeping, TTFL, circadian rhythm

## Abstract

The recently observed circadian oscillations of the intestinal microbiota underscore the profound nature of the human–microbiome relationship and its importance for health. Together with the discovery of circadian clocks in non-photosynthetic gut bacteria and circadian rhythms in anucleated cells, these findings have indicated the possibility that virtually all microorganisms may possess functional biological clocks. However, they have also raised many essential questions concerning the fundamentals of biological timekeeping, its evolution, and its origin. This narrative review provides a comprehensive overview of the recent literature in molecular chronobiology, aiming to bring together the latest evidence on the structure and mechanisms driving microbial biological clocks while pointing to potential applications of this knowledge in medicine. Moreover, it discusses the latest hypotheses regarding the evolution of timing mechanisms and describes the functions of peroxiredoxins in cells and their contribution to the cellular clockwork. The diversity of biological clocks among various human-associated microorganisms and the role of transcriptional and post-translational timekeeping mechanisms are also addressed. Finally, recent evidence on metabolic oscillators and host–microbiome communication is presented.

## 1. Introduction

From the moment of birth, a child is immediately exposed to a vast array of microorganisms that gradually colonize the body. Besides potentially dangerous pathogens, many microbes have co-evolved with humans over millions of years, building a symbiotic relationship that became a prerequisite for a healthy life [1,2,3,4].

These microorganisms represent virtually all branches of life, including most dominant bacteria, as well as archaea, fungi, algae, and small protists, collectively referred to as the microbiota [5,6,7,8]. As microbiota settle in, they lay the foundations for a microbiome, shape the immune system, and form a close relationship with various other body systems and organs, strongly influencing their functions [3,4,9]. The concept of the microbiome is more comprehensive, capturing not just the microorganisms but also their “theatre of activity”, including all the metabolites being produced, the microbial genome, and other environmental conditions [7,8].

Intriguingly, the microbiome represents a dynamic and interactive micro-ecosystem susceptible to variations in both temporal and spatial dimensions [7]. Emerging observations from the past decade have demonstrated the circadian rhythmicity of the gut microbiome’s composition, metabolic activity, function, distribution, total biomass, and even its interactions with the host [10,11,12,13,14,15,16]. Studies indicate that in terms of composition, 60% of the total intestinal microbiome displays circadian oscillations, which in mice and humans are attributed to 10–20% of the commensal bacterial taxonomic units, affecting around 20% of microbial functional pathways [12,14,15].

The rhythmic relationship between the microbiome and the host is bidirectional. The host’s functional circadian clock is crucial for maintaining microbial rhythmicity, providing time cues, and synchronizing microbiome activity with body physiology [10,11,12,17,18,19]. On the other hand, microbial oscillations can influence the host’s physiology, gene expression, brain functions, circadian rhythm, immune system, and disease susceptibility [10,15,19,20,21,22,23,24,25]. Microbiota activity can induce diurnal metabolic rhythms in the cells lining the intestinal wall [24], and the depletion of the microbes disrupts the circadian clocks of intestinal epithelial cells [21,26]. Therefore, the disturbances in normal microbial oscillations might contribute to the development of various medical conditions, including metabolic syndrome, obesity, diabetes, and cardiovascular diseases [23,27,28,29,30].

All of this underscores the importance of the rhythmic relationship between humans and their tiny inhabitants. However, where does the rhythmicity of the microbiome originate from? Is it purely a consequence of the host’s circadian clock activity? Emerging evidence indicates that at least some microbiota species have intrinsic timing mechanisms that empower them with an ability to anticipate and adapt to the ever-changing environment of the gastrointestinal tract [31,32,33]. This important finding has raised many pivotal questions, the answers to which may be crucial in understanding the intricacies of biological timekeeping and could potentially revolutionize various branches of medicine.

How many other human-associated microorganisms possess biological clocks?Is there a single, conserved primordial timekeeping mechanism, or has evolution given rise to diverse proto-circadian systems?What constitutes the core of cellular timing systems? Is traditional self-sustainability necessary, or can adaptive advantages be achieved with simpler mechanisms?How do microbes track the time inside the human intestine in the absence of light cues?

However, answering these questions necessitates a profound understanding of the evolution of biological clocks. To address these issues, this review explores the fundamentals of the rhythmic cooperation between humans and microbiota by tracing back to the origins of timing systems. Set within an evolutionary context, the latest advancements in the field of cellular chronobiology are discussed in detail, highlighting the potential applications of this novel knowledge in medicine, especially those associated with human-related microbes.

## 2. The Clock That Ticks the Life Rhythm

While fossil records indicate that the earliest life on Earth originated 3.5–3.7 billion years ago [34,35,36,37], ongoing analyses suggest an alternative scenario where the Last Universal Common Ancestor (LUCA) might have emerged even earlier, predating 4 billion years ago [38,39,40]. Since then, primitive life has been gradually evolving to the rhythm dictated by the alternating cycle of day and night. The timing of sunrise and sunset played such a significant role in the functioning of the first organisms that their internal physiology and external behavior were compelled to adapt to it [41,42,43,44,45]. This interaction between biological systems and the changing intensity of sunlight turned into an enormous advantage, a fundamental evolutionary feature known as circadian rhythm (CR).

CRs are defined as autonomous and self-sustaining oscillations that regulate and synchronize physiological processes in a 24 h cycle while also adjusting to zeitgebers (external environmental time cues capable of entraining CRs, such as light and temperature) [46,47]. CR became a necessity for early organisms and was one of the very first demonstrations of environmental evolutionary adaptation. Therefore, CR is deeply embedded in the functioning of every living organism [48,49,50]. CRs manifest themselves in a multitude of ways, ranging from barely perceptible to strikingly evident patterns [47]. Circadian regulation governs all vital processes taking place in the organism, beginning with the most essential ones, such as metabolism modulation and gene expression, continuing with cell division, immune system regulation, and homeostasis maintenance, culminating in shaping behavioral actions, inter-organism interactions, and even reproduction [45,51,52,53,54].

What distinguishes CR from diurnal rhythms, which are general biological processes occurring in a 24 h manner, is that CR is naturally internal and endogenous. CR can persist and maintain its rhythm even in an environment that lacks zeitgebers. While diurnal rhythms are driven solely by external time cues, CR has its own timekeeping system, a biological clock called the circadian clock (CC) [46,47]. CC operates based on endogenous cellular mechanisms that can autonomously measure 24 h, empowering organisms with the ability to anticipate day–night cycles [51,55,56]. CC provides internal temporal organization and ensures the proper timing between environmental phenomena and an animal’s behavior, allowing organisms to occupy a niche based on time rather than exclusively on physical space [45,46]. The widespread presence of CCs across various levels of organization and complexity, in myriad organisms, and within various ecosystems underscores their importance in conferring adaptive advantages [42,48,57,58].

## 3. The Expanding World of Bacterial Biological Clocks

CR has ancient evolutionary roots, extending back several billion years. A growing body of research provides evidence that evolutionary pressures and natural selection have been favoring the formation of genetic circuits responsible for tracking daily fluctuations of light and temperature already in early prokaryotes [50,59,60,61]. In blue-green algae, specifically the cyanobacterium *Synechococcus elongatus* PCC 7942, the CC has been discovered to regulate the expression of various genes and proteins, which in turn control numerous biological processes such as nitrogen fixation, carbon metabolism, chromosome compaction, cell division, global gene expression, amino acid uptake, and photosynthesis [21,61,62,63,64,65,66,67,68,69,70,71].

Recent findings indicate that the *kaiA* gene, an integral component of the central oscillator that underlies the circadian system in cyanobacteria alongside *kaiB* and *kaiC*, has been present in most cyanobacteria since their origin approximately 3000 ± 500 million years ago [72]. It is noteworthy that the interplay between various CC components creates not just a linear circadian model but a complex network of interactions and biochemical oscillations, which give rise to a well-defined CR in cyanobacteria [50,61,64,73]. This constitutes one of the most well-researched circadian systems across all domains of life [74] with fully robust, temperature-compensated, and self-entrained CC [75].

It is well established that the CRs in cyanobacteria govern cellular fitness in cyclic environments, providing an evolutionary advantage over cyanobacteria cultures with a mismatched circadian period [42,76,77,78]. A notable example of such an advantage is the fact that CR enables cyanobacteria to anticipate and prepare for dark-induced resource limitations, which is a key challenge faced by these organisms [64,79]. Conversely, arrhythmic counterparts seem to fare better in constant light conditions. This suggests that the benefits associated with a CC are context-specific and may not inherently confer advantages to the organism’s physiology under vastly altered or artificial conditions [42,73,78].

Until recently, cyanobacteria were widely believed to be the simplest organisms with a CR [50,59,61]. However, emerging evidence indicates the presence of CR in other prokaryotic organisms. For example, circadian-like properties were observed in two bacterial species: *Bacillus subtilis* [32,80] and *Klebsiella aerogenes* [33,81,82]. These discoveries could be considered groundbreaking, as scientists have found functional CCs in prokaryotes outside the phylum Cyanobacteria. This represents a significant development in the understanding of the origins of CR, as it can confirm previous suggestions that non-photosynthetic and human-related bacteria may also be capable of exhibiting a defined CR [74]. Since *K. aerogenes* and *B. subtilis* are frequently found in the gastrointestinal tract of many individuals, this raises the possibility of mutual communication between microbial CCs and the cells of the human gut epithelium.

*K. aerogenes* exhibits endogenously generated and robust, temperature-compensated CR of the MotA motor protein expression, which is displayed in the form of rhythmic swarming motility patterns [33,83]. Interestingly, the bacterial CC is particularly sensitive to an important human-derived signaling molecule, melatonin, which is a pineal and gastrointestinal hormone. Melatonin has been observed to not only substantially enhance the swarming activity over a 24 h cycle but also to aid in the synchronization of growing bacterial cultures [33,83]. Moreover, melatonin affects a wide range of growth-specific genes that regulate outer membrane, periplasmic, and cytoplasmic proteins. These proteins play critical roles in various cellular processes, including pilus biosynthesis, biofilm formation, stress response, carbohydrate transport, and metal ion homeostasis. Therefore, melatonin enhances bacterial attachment to the gut epithelium and improves colonization of the host [81]. Furthermore, the CC of *K. aerogenes* can entrain variations in ambient temperature, which correspond to changes in the host’s body temperature [82].

In the case of *B. subtilis*, researchers have discovered CR in the expression of the gene encoding a blue light photoreceptor. The observed oscillations had a free-running period, temperature compensation, and the capacity to entrain changes in light and temperature, indicating the presence of a fully functional CC [32]. Intriguingly, the presence of glucose in the media appears to favor the establishment of free-running rhythms under light/dark cycles but inhibits such rhythms under temperature cycles. This is not the first observation of circadian behaviors in this bacterial species. Several earlier reports have documented 24 h oscillations in gene expression, differentiation, and spore formation in *B. subtilis* [74].

These findings imply that the CC present in *K. aerogenes* and *B. subtilis* might allow for crosstalk between microbial and human timekeeping systems [32,33,81]. Environmental factors such as temperature fluctuations, hormone levels, and nutrient availability may serve as zeitgebers for the microbiota’s timekeeping mechanisms [32,82]. Gaining a more in-depth understanding of, and ability to manipulate, the CR of specific human microbiota species could have substantial health implications and may pave the way for novel therapeutic approaches.

In recent years, there has been increasing interest in employing *Bacillus* species as probiotics owing to their unique characteristics and high safety profiles [84]. Probiotics containing *B. subtilis* can be used effectively in treating gastrointestinal disorders and symptoms of irritable bowel syndrome [84] and may also be beneficial in preventing certain infections through mechanisms such as direct competition with pathogens [85,86] or immune system stimulation [87]. In contrast, *K. aerogenes*, despite being a commensal gut bacterium harmless to healthy individuals, can become a lethal opportunistic pathogen in immunocompromised and hospitalized patients, leading to infections of the urinary and respiratory tracts, also demonstrating antibiotic resistance properties [88,89]. Understanding that melatonin affects the pathogenicity of *K. aerogenes* through interaction with its biological clock and considering that this hormone is a crucial biomarker for circadian dysregulation [90] could be leveraged effectively.

Disrupting the pathogen’s CRs could alter its gene expression, impair its ability to anticipate environmental changes, decrease swarming, and hinder its ability to efficiently acquire nutrients, which could slow down its growth rate and reduce pathogenicity. This could neutralize the advantages gained by the pathogen’s timekeeping abilities and diminish its competitiveness with other members of the healthy microbiome. On the other hand, targeting the CCs of commensal microbes could preserve and enhance the protective role of the healthy microbiome. The capability to influence the physiology of bacteria associated with human health through the modulation or entrainment of their CCs, potentially through the host’s CRs, could be invaluable in numerous medical applications.

Even more promising findings are emerging about the timekeeping abilities of human-associated bacteria. *Escherichia coli*, despite extensive research on its pathogenic and antibiotic-resistant strains, is predominantly a non-pathogenic bacteria found in the gut microbiome of almost all individuals [91,92,93,94]. And although *E. coli* is frequently referred to as a “commensal” bacterium, cumulative evidence shows that its relationship with humans should be more accurately described as mutualistic [95]. *E. coli* is responsible for synthesizing essential vitamins such as K and B12 [91,96,97]. It is also one of the first bacteria to colonize neonates at birth [2,98,99], providing a safe, anaerobic environment for other microbiota species [2,95]. Additionally, *E. coli* is known to exert a protective effect by preventing the colonization of pathogens [93].

Historically, *E. coli* was one of the first bacteria that scientists initially suspected to be a potential candidate for exhibiting circadian-like cycles [100]. Research dating back nearly a hundred years indicated the presence of ultradian rhythms (biological rhythms that occur within a period shorter than 24 h) in its growth rate [101]. Later, it was determined that these rhythms follow a 21 h cycle [102]. Despite this, the absence of temperature compensation and entrainment prevented the confirmation of the observed phenomena as definitive CR.

Intriguingly, when supplemented with a transplanted KaiABC cyanobacterial complex, *E. coli* exhibits circadian oscillations of KaiC phosphorylation, which can drive transcriptional output for three days with a period matching the day–night cycle [103]. According to the authors of the study, implementing such bioengineering methods may provide a pathway to modulate the CRs of gut microbes. These kinds of innovative approaches offer a range of chronotherapeutic applications, effectively addressing issues like jet lag-associated dysbiosis and optimizing the management of circadian rhythm disorders. Such solutions could also be incorporated into a broader biotechnological context, encompassing automated drug administration and circadian control in industrial microbial processes [103].

Given that *E. coli* is metabolically capable of supporting a heterologous clock, it harbors biochemical machinery capable of generating biological rhythms. Recently, researchers identified a potential homologous gene, *radA*, to the *kaiC* clock gene in *E. coli* [104]. Observations showed that the expression of *radA*, along with other genes involved in various cellular functions, displays circadian periodicity, and the rhythm persists over time. However, the authors did not demonstrate temperature entrainment and compensation [104]. Since the genetic background of *E. coli* can influence mammalian CC genes, it can be inferred that both pathogenic and commensal strains may affect the host’s immune system CR through their rhythms [105]. It is well established that CR controls the immune response to infection in eukaryotic hosts [106]. Hence, by adjusting the internal physiology, bacterial clocks could help them adapt to a difficult environment altered by antibiotics and the burst of antibodies. The discovery of circadian rhythm-driven molecular clock gene expression in *E. coli* has the potential to revolutionize antibacterial therapies for infectious diseases [104].

Alongside pathogenic *E. coli*, *Klebsiella pneumoniae* is one of the pathogens responsible for the highest number of deaths attributable to antibiotic resistance [107]. Apart from its dangerous pathogenic properties, *K. pneumoniae* has also recently been recognized as a prevalent commensal gut bacterium [108]. It can frequently be found in the healthy human microbiota of the skin, oral cavity, and respiratory and gastrointestinal tracts. Unfortunately, these areas also serve as reservoirs for many opportunistic pathogenic strains [109,110,111]. *K. pneumoniae* is a common cause of various diseases such as pneumonia, urinary tract infections, meningitis, sepsis, and intra-abdominal infections, especially among newborns, the elderly, or immunocompromised individuals [110,111].

A study conducted five decades ago hinted at the presence of certain rhythms in *K. pneumoniae* [112]. However, the research failed to demonstrate conclusive evidence or defining characteristics (temperature compensation and entrainment) to support the existence of CC [59]. While it remains unconfirmed whether *K. pneumoniae* possesses a robust and persistent CR, there is indeed a rhythm affecting its growth. Perhaps leveraging the mechanisms underlying these rhythms may hold the potential for uncovering innovative therapeutic alternatives that differ from antibiotic treatment.

Addressing this topic is of utmost importance due to the significant challenges involved in treating infections caused by *K. pneumoniae*. This is primarily attributed to the restricted range of antibiotics that effectively target this pathogen [113]. Additionally, despite the extensive use of available antimicrobial therapies, the mortality rate associated with this bacterium remains alarmingly high [114]. A striking example of such a situation occurred in 2016 when a woman infected with a specific, pan-resistant strain of *K. pneumoniae* died due to septic shock. According to the subsequent analysis, the isolate demonstrated resistance to all 26 available antimicrobial drugs [115]. Cases like these are extremely rare, but they may become more frequent over time, underscoring the importance of finding novel solutions regarding the Antibiotic Resistance Crisis [116].

Taken together, it can be observed that numerous bacteria belonging to various phyla are likely to regulate their physiology rhythmically, following circadian-like or ultradian periods. Currently, it is known that the misuse and overuse of antibiotics have given rise to a significant global healthcare challenge of antimicrobial resistance, leading to the widespread occurrence of severe infections, prolonged hospitalization, and higher mortality rates [116,117]. A deeper understanding of the mechanisms governing bacterial biological clocks, regardless of their level of complexity, could enable researchers to impede the development of bacterial escape mechanisms, delay the emergence of resistance to specific antibiotics, and thereby affect the effectiveness of treatment. This could be achieved, for instance, by manipulating bacterial biological clocks to disrupt their normal physiology and coordinating with antibiotic administration at specific times of day when the pathogen is in a weakened state [74].

## 4. A Missing Line on the Evolutionary Map: Eukarya–Archaea

The identification of functional timekeeping mechanisms in Archaea, a lineage tracing back to the dawn of life on Earth, is a key missing piece in understanding the origins of CR. However, it is also of significant importance to discern the evolutionary history of Archaea and its relation to Eukarya, which is certainly one of the most contentious topics in the discussion of life’s evolution [118]. Notwithstanding, the dominant phylogenetic classification system for the past three decades has been based on three domains of life (Figure 1, 3D); very recently, an expanding body of research [119,120,121] has reignited the debate regarding the adoption of the previously proposed two-domain tree of life model (Figure 1, 2D), [122,123,124].

Archaea exhibit characteristics that resemble both bacteria and eukaryotes. Specifically, archaea share some similarities with bacteria in terms of external appearance; both are prokaryotes lacking a nucleus and other membrane-bound organelles, and they possess circular chromosomes and reproduce by binary fission [125,126]. However, those two domains differ substantially in terms of internal morphology and functionality. Certain archaeal species exhibit close resemblance to eukaryotic cells in genetic information-processing machinery; they encode similar proteins (that were previously thought to exist only in eukaryotes) and show several molecular features unique to eukaryotes [120,124,126,127,128]. Although the debate on this matter remains intense [119,129,130,131,132,133], advances in phylogenomic, molecular, and genetic studies, as well as the discovery of numerous new archaeal species, including those believed to be precursors of the initial eukaryotic cells, provide compelling evidence supporting the topology of a two-domain Tree of Life [120,121,127,134,135,136,137,138,139].

Consequently, Eukarya may no longer be considered a separate domain but rather as originating from within an archaeal branch [124,126,128,133,134,136,137]. Eukaryotes could also be viewed as “chimeras”, whose genomes have gradually evolved by acquiring a range of bacterial and archaeal genes through the process of horizontal gene transfer (Figure 1, CHIMERA) [126,140]. Likewise, inasmuch Archaea seem to stand “between” Eukarya and Bacteria, it is plausible that archaeal timekeeping mechanisms might be hybrids, incorporating features observed in both bacterial and eukaryotic cells [141]. Therefore, it could be argued that Archaea served as precursors of timekeeping and more research on them might elucidate the subsequent evolution of the diversity in forms and functions of timing systems across nucleated and anucleated cells [142].

Undoubtedly, the twenty-first century marks a pivotal era for microbiology, with novel findings related to the Tree of Life emerging at an unprecedented rate [118]. Further investigations into the origins of life hold the potential to open new scientific avenues, which could foster significant advancements in the field of evolutionary research shortly [120]. Understanding the intricate relationship between Archaea and Eukarya seems to be the crucial piece in comprehending the evolutionary trajectory of life and the beginnings of the first CRs and primordial biological clocks.

## 5. Subtle Traces of Archaeal Timekeeping Capabilities

In the context of the microbiome, archaea have been studied less than bacteria. Nonetheless, archaea have been demonstrated to be a stable part of human-associated microbiota, with estimates suggesting they may account for up to 20% of microbiota species [143,144]. Through syntrophic processes, which involve collaboration with other microorganisms, archaea assist in breaking down complex organic compounds that are indigestible by humans alone [145]. Certain archaea species are known to be involved in carbohydrate metabolism, and some can metabolize methane, thereby reducing hydrogen levels and promoting the growth of saccharolytic bacteria [146]. These are known as methanogens and represent the most dominant archaea in the human gut. Members of the Haloarchaea class also contribute to the diversity of the human microbiota [147].

Methanogens may have a beneficial impact on the body as they can support digestion and absorption of nutrients, modulate immune system homeostasis, and maintain a healthy balance of gut microbiota [148]. On the other hand, recent research suggests that several archaeal species may be involved in the development of systemic diseases [147,148], and the role of gut methanogens in contributing to pathological conditions has been underestimated [147]. They could be associated with a range of conditions, including severe colon diseases, obesity, irritable bowel syndrome, periodontal and endodontic diseases, inflammatory bowel disease, and even atherosclerosis [147,148].

Currently, traces of CR within the Archaea domain have been identified in only two species from the Haloarchaea class, neither of which is related to the human microbiome [141,142]. Notably, under nutrient and oxygen-limited conditions, these microorganisms indeed exhibit cyclic anticipatory behaviors like those regulated by a defined CC. In the species *Halobacterium salinarum*, diurnally entrained oscillatory transcription was observed in approximately 12% of all genes, manifesting as free-running rhythms that persisted over three days in constant darkness following entrainment with light–dark cycles [142]. Interestingly, even though the microbes were maintained in conditions with constant oxygen levels, it was found that a substantial part of the cycling genes were associated with the regulation of and response to changes in oxygen concentrations. This implies the existence of an unidentified timekeeping mechanism that allows *H. salinarum* to anticipate and adapt to fluctuating oxygen levels throughout the day. Remarkably, due to oscillatory gene transcription, the physiology of this archaeon can toggle between oxic and anoxic states and split the degree of induction of many important metabolic processes between dark and light phases. This highlights the significance of this rhythmic behavior in optimizing resource utilization under cyclic light–dark conditions [142].

Regarding the second archaeon with observed circadian properties, scientists have successfully identified four homologs of cyanobacterial *kaiC* in its genome, termed *cirA*, *cirB*, *cirC*, and *cirD*. Significantly, *Haloferax volcanii* displayed diurnally synchronized oscillations in the expression of *cir* genes under 12 h light/12 h dark cycles [141]. Through gene knockout experiments, it was demonstrated that each of these genes is essential for maintaining this light-driven rhythm. However, although these diurnal responses persist without light in both cases, they cannot be defined as CRs. The reason for this is that other defining properties of CC, such as temperature compensation, have not been recognized yet and need further investigation [60,141].

## 6. Delving Deeper into Prokaryotic Oscillators Unveils the Landscape of Timing Systems

Despite the ambiguity surrounding the origins of Archaea [149,150,151,152], these prokaryotes probably have coexisted with cyanobacteria, implying similar environmental pressures and cross-domain interactions that shaped their evolution [141]. This could potentially explain why scientists have identified many KaiC versions both in archaeal and cyanobacterial species [153,154,155,156].

Excluding KaiC in cyanobacteria, the function of proteins belonging to this class is largely unknown [154]. In *Legionella pneumophila*, which is an opportunistic human pathogen, a KaiC homolog enhances the fitness of the bacterium by increasing its resistance to oxidative and ionic stress, notwithstanding its lack of interaction with the KaiB homolog. Nevertheless, it seems that the *kaiC* homolog of *L. pneumophila* does not appear to encode any CC and does not regulate CR [157]. In archaea, for example, KaiC-like proteins were found to undergo autophosphorylation, be crucial for motility, and have a central role within signal transduction mechanisms [158,159].

As *kaiC* homologs are present in multiple copies in most archaea but not universally in bacteria, it is plausible that KaiC-like proteins initially emerged in archaea and were subsequently transferred to cyanobacteria [100,154,155,158], though counterarguments to this thesis exist [160]. What is known for sure is the fact that many diverse evolutionary events, such as horizontal gene transfer, mutations, or rearrangements, have contributed to the formation of both archaeal and bacterial genomes [161]. The likelihood that other archaea possess CCs akin to those in photosynthetic bacteria is negligible [158], particularly given that other cyanobacterial species possess different timekeeping mechanisms compared to the model *S. elongatus* [154]. However, there are indications that hint at the characteristics of a potential ancestral timing system [158].

Since certain prokaryotes possess homologs of the *kaiB* and *kaiC* genes, it is plausible that a rudimentary form of the biological clock might have evolved without the *kaiA* gene, oscillating in a damped (rhythms fade over time in constant conditions) or hourglass-like pattern (resembling a stopwatch requiring constant resets), yet still synchronizing and responding to environmental cycles (Figure 2) [162]. Interestingly, the components of this hypothetical basic biological clock are present in some archaeal methanogens [142,158], which are believed to have existed on Earth for approximately 3.4–2.6 billion years. This time frame coincides with the evolution of cyanobacteria, suggesting that these prokaryotes were likely subjected to similar selective pressures that influenced their primordial timekeeping systems [141,163]. However, as exemplified by the case of *L. pneumophila*, a *kaiBC*-based system in methanogenic archaea may not be associated with circadian function [158].

Importantly, *kaiA* is considered an evolutionary innovation that emerged to accomplish robust CR [164,165]. Among bacteria, only a subset of cyanobacteria species possesses this gene [154]. In contrast, genome analysis reveals the presence of *kaiB* and *kaiC* in a wide range of bacterial families across different phyla [166]. Species such as *Rhodobacter sphaeroides* [167], *Rhodopseudomonas palustris* [165], and the marine cyanobacterial genus *Prochlorococcus* [73,168] exhibit circadian-like pattern oscillations. Notably, *Prochlorococcus*, despite being the most abundant of all cyanobacteria, does not possess the *kaiA* gene [169]. And although the timing systems of all these bacteria cannot be classified definitively as CCs, they do confer some characteristics and advantages of an autonomous oscillator. Increased fitness in 24 h cyclic environments, observable even in the absence of rhythms under constant conditions, represents a type of KaiBC-based minimal circadian system, or a proto-circadian clock, which likely served as one of the evolutionary prototypes on the path to the emergence of self-sustained oscillators [164,165,167,170].

Furthermore, it is conceivable that a most primitive biological timer might require only a single core protein, likely KaiC, along with a few additional components. Such a system should be capable of driving diurnal rhythms, just as in the case of the archaeon *H. volcanii* [141], and its entrainment could be achieved by utilizing the ATP/ADP ratio [154]. The emergence of the *kaiC* gene is estimated to have occurred sometime between the LUCA and the origin of cyanobacteria. Thus, the original *kaiC* gene, or its precursor, was probably already present in LUCA [60], supporting the hypothesis that first biological clocks may have evolved from the most basic hourglass timers or through damped oscillators, ultimately into self-sustained oscillators [164,171]. The archaeon *H. salinarum* is a good example of an organism that possesses a primordial KaiC-based hourglass timer, which proves sufficient in more regular environments [142,164].

Nonetheless, it is worth emphasizing that the fundamental mechanism of cyanobacterial CC is not primarily centered around genetic regulation. Rather, the post-translational modifications play a critical role in prokaryotic clockwork. This could suggest that even in eukaryotes, the core circadian timekeeping is not necessarily embedded within genetic activity but might originate from more ancient, yet unidentified, cellular mechanisms [56].

Beyond the well-defined CCs, there exists a wide variety of biological clocks in different organisms that have distinct functions, mechanisms, and characteristics, which include timing system, timekeeping system, circadian clock, ultradian clock, damped oscillator, and hourglass timer (Figure 2).

Biological clocks are systems in living organisms that allow them to respond to the passage of time, anticipate environmental changes, and regulate and coordinate various physiological processes. This umbrella term includes all mechanisms, structures, and pathways involved in timekeeping or time-sensing, from the cellular to systemic levels.

The timing system and its mechanisms refer to precise tools, primarily at the cellular or molecular level, that enable organisms to sense, be influenced by, and respond to time changes. Various timing mechanisms can function within one general timing system.

The timekeeping system and its mechanisms refer to the cellular machinery used by organisms to actively track and measure time. Different timekeeping mechanisms can function within a single timekeeping system, even alongside other timing mechanisms. It could be considered a subset of or a more sophisticated timing mechanism/system, distinguishing itself by not being solely passive and subject to time change.

The circadian clock is the most specialized form of biological clock, self-sustaining and operating on a nearly 24 h cycle. Notable for temperature compensation, robustness against disturbances, and maintenance of a stable period even in constant conditions.

The ultradian clock, as observed in baker’s yeast, operates at sub-circadian time intervals but still exhibits self-sustainability.

A damped oscillator is a timekeeper that loses rhythm over time in constant conditions. Though not fully autonomous, it can perform comparably to a circadian clock under rhythmic conditions. It can be compared to a pendulum moving against resistance forces, causing it to lose energy over time. To work perfectly, it requires the right conditions.

An hourglass timer is a rudimentary timing system analogous to a stopwatch. It operates for a specific duration before being reset by a rhythmic environment, much like how the sand in an hourglass runs for a set duration before being flipped.

## 7. A Fork in the Road of Circadian Rhythms—TTFL and PTO Mechanisms

The transcription–translation feedback loop (TTFL) appears to be a fundamental oscillation mechanism driving CRs across eukaryotes, including plants [172,173], fungi [174], and animals [175]. This autoregulatory negative feedback loop operates through the transcriptional regulation of clock genes, which encode both positive and negative regulatory parts of the system [176]. Its protein components form complex interactions, driving internal rhythmicity, which in turn influences various global physiological and behavioral processes of the organism, allowing it to anticipate and adapt to environmental changes [56,177].

Although the general TTFL mechanism is conserved across different phylogenetic kingdoms, the principal genetic components, or “gears”, differ markedly and exhibit minimal homology [49,178]. The sequences of the core clock genes of animals, fungi, and especially plants are just dissimilar [179,180,181]. Even within the animal kingdom, although *Drosophila* and mammals possess homologous clock proteins, the interactions, regulation, and functions of these proteins differ [51]. There is also no homology between the core clock components of cyanobacteria and eukaryotes [49]. Moreover, homologs of the cyanobacteria gene cluster *kaiABC* have been found in only a few other bacteria [165]. Neither the recently discovered *B. subtilis* nor *K. aerogenes* encode *kaiABC* sequences in their genome. In the case of *K. aerogenes*, even though one of its proteins exhibits some homology with cyanobacterial KaiC, motif analyses did not substantiate the hypothesis that this bacterium employs a KaiC-driven timekeeping mechanism [32,33].

Consequently, as the structure and organization of core clock genes vary significantly between species, it is evident that they lack phylogenetic continuity across phyla [182]. This observation leads to the inference that various biological clocks had independent evolutionary origins [179,180]. Notwithstanding these structural and evolutionary distinctions, timekeeping mechanisms across different organisms had to follow a path shaped by common evolutionary pressures. Responsive adaptation to seasonal and diurnal environmental fluctuations in light and temperature was simply a feature that could not be missed [45]. Therefore, it is reasonable to think that, in addition to genetic-based clocks, there should have emerged timing systems that extend beyond the TTFL model and still retain the capability to sustain rhythmicity.

In the initial phase of research into CC in cyanobacteria, the *kaiABC* gene cluster was believed to function exclusively as a simplified TTFL model [183,184]. Later, it was found that post-translational modifications, particularly the phosphorylation cycle of KaiC protein, play a crucial role in regulating CR in cyanobacteria [184,185,186,187] and that the genetic autoregulatory feedback loop is not mandatory for maintaining CR [188]. Further studies, including the successful recreation of the KaiC phosphorylation rhythm in vitro, achieved solely by combining KaiA, KaiB, and KaiC in the presence of ATP [186], broadened the scope of research on biological clocks as scientists realized that genetic circuits are dispensable for generating circadian oscillations. These revelations clarified that the primary timekeeping mechanism in cyanobacteria operates based on a purely post-translational oscillation mechanism (PTO) [178,186,189].

A sudden paradigm shift in the understanding of possible clock mechanisms paved the way for the discovery of CR in eukaryotic, naturally anucleate cells—red blood cells of mice and humans [190]. Importantly, the emergence of CR was linked to the activity of peroxiredoxin proteins, the second most abundant proteins in erythrocytes [190,191]. Such a meticulous investigation into PTO in eukaryotes was indeed a monumental scientific breakthrough. It revealed not only that the presence of a nucleus is not a prerequisite for eukaryotic cells to exhibit CR but also that they may possess alternative mechanisms to keep track of time beyond relying solely on transcription-based oscillators [49,192].

Altogether, these findings shattered the previously held dogmatic view that clockwork and circadian regulation are based entirely on TTFL [56,178].

## 8. Why Has Evolution Decided to Start Measuring Time?

Peroxiredoxins (PRDX) are a family of antioxidant enzymes responsible for catalyzing the reduction in metabolic toxic byproducts collectively named reactive oxygen species (ROS) [193,194,195,196]. PRDX are widespread and present in virtually all living organisms [197,198,199,200]. Neutralizing ROS, i.e., by reducing hydrogen peroxide (H_2_O_2_) to water, is how PRDX help maintain redox homeostasis, minimize oxidative stress, and thereby counteract related cellular damage [201]. Thus, PRDX are acknowledged as crucial participants in defending cells against the deleterious effects of oxidative stress [195].

The cyclical oxidation–reduction reactions of PRDX occur rhythmically and are overlaid by circadian oscillations. These have been shown to meet all defining properties of CR, representing potentially fully functional CC [178,190,191]. Notably, PRDX redox oscillations are observed in every living organism, spanning from unicellular species to the most advanced mammals [191], and have even been detected in archaea [202]. This suggests that potentially every microbiota species may possess some form of timing mechanism.

The occurrence of the PRDX redox rhythms in cyanobacteria does not require the *kaiA* gene [202,203]. While the evolution of TTFL has been divergent in different organisms, PRDX oscillations are phylogenetically conserved across all domains of life, serving as a potential universal timekeeping system [202]. Thus, it could be assumed that PRDX probably played a pivotal role in the evolution of the first biological clocks [49,204]. Moreover, it is tempting to propose that this mechanism might represent the original pacemaker, the first ancestral clock [185,202].

Such a scenario may be consistent with the selective stress hypothesis regarding the origins of the biological clocks. Accordingly, a PTO-based timing mechanism that could concurrently utilize the activity of proteins associated with antioxidant function would have been highly advantageous during and after the Great Oxygenation Event, a period of dramatic change for early life forms [49,178,204].

The evolution of photosynthesis in cyanobacteria played a significant role in the transition of the Earth’s atmosphere from a reductive to an oxidative state [205,206]. This rapid increase in oxygen levels made oxidation an exceptional environmental stressor, which created an evolutionary pressure for the development of a metabolic mechanism capable of defending cells from oxidative stress. Life forms that were able to survive in the new oxygen-rich environment were those capable of both metabolizing oxygen and effectively detoxifying ROS [204]. However, the presence of PRDX may have not only empowered cells with antioxidant capabilities but also provided a foundation for the first, most basic biological clocks [202]. Given that the solar cycle modulates the oscillations of oxygen consumption and ROS production, an integrated endogenous antioxidant system could broaden the scope of its operation by additionally enabling organisms to anticipate environmental changes. PRDX seem to be well-suited to serve both these functions, as they are sensitive to the rhythmic fluctuations of ROS, which are driven by the alternating oxidative states due to photosynthesis during the day and the contrasting hypoxic conditions at night [207].

Another intriguing theory that addresses the emergence of circadian regulation is the “Escape from Light” hypothesis, which suggests that biological clocks evolved as a response to the daily fluctuations in light intensity [208,209,210]. In the initial stages of evolution, the atmosphere of primitive Earth was quite thin, devoid of oxygen and ozone. Without an effective protective layer, primordial organisms were highly vulnerable to the damaging effects of solar radiation, particularly ultraviolet (UV) radiation. It is estimated that the intensity of UV radiation was around 3–4 orders of magnitude higher in the ocean’s surface layer compared to present-day levels [211]. Such extremely high radiation was very destructive to DNA even underwater, as UV radiation can penetrate considerable depths due to its short wavelength. To counteract this, early life forms developed an array of defensive mechanisms and structures [212]. As a result, specialized blue light photoreceptors called cryptochromes and proto-biological clocks have evolved from photolyases (proteins primarily involved in DNA repair) to protect cells from UV radiation damage and enable them to sense and anticipate daily time changes [213,214]. These early light-sensing systems endowed organisms with an invaluable ability to predict and avoid, or at least mitigate, the harmful effects of intense light exposure. The emergence of the *kaiC* gene further supports this hypothesis, as the protein it encodes might have originally been associated with DNA replication, DNA damage repair, and/or RNA metabolism [60].

Currently, it is known that the circadian control of UV resistance is a feature present in both eukaryotic and prokaryotic species [215]. Interestingly, it has been suggested that the “DNA damage” alone, via the activity of TTFL components, could constitute a zeitgeber for both circadian and metabolic cycles in eukaryotic organisms absent of photoreceptors [216].

Certainly, maintaining a protective mechanism against light damage constantly throughout 24 h is not energy efficient. Consequently, it was a natural progression in evolution to develop some form of clock system. Biological clocks facilitate the conservation of energy normally used for protective mechanisms during safe, low-light periods, simultaneously enabling critical biological processes to intensify without the risk of UV radiation damage [215]. For instance, the single-celled alga *Chlamydomonas reinhardtii* displays the highest vulnerability to UV radiation during mitosis, with peak sensitivity being observed at the end of both day and night periods [210]. On the other hand, processes that benefit from intense radiation can be initiated in advance to improve their efficiency. This is the case in plants, which maximize the potential of their photosynthesis by responding anticipatorily to variations in daily light cycles [58].

Notably, the process of photosynthesis is inherently associated with an increase in oxidative stress, and PRDX appear as crucial agents in protecting chloroplasts from this light-associated damage [217,218]. Therefore, a second, more ancestral “cog” of the circadian system in the form of the PRDX-based oscillations appears to be an ideal partner for the transcription–translation-based system, which is vulnerable to stress factors. Nevertheless, there is still no clear consensus regarding the mechanism behind PRDX oscillations. What drives the circadian timing of redox balance is a very controversial topic [49,192,219,220,221].

## 9. Bridging Redox Metabolism, Cellular Communication, and Antioxidant Function

Certainly, the evolution of biological clocks was not motivated solely by the impact of external environmental factors. The temporal regulation of internal structure, organization, and metabolism also plays a significant role, conferring substantial adaptive advantages [49]. In the context of the PTO mechanism, the physiological significance of PRDX is supported by the fact that this family of proteins is ubiquitous in all known species [201], including both eukaryotic [222,223,224] and prokaryotic [202,225,226], with a single exception of one bacterial species that is incidentally susceptible to being killed by H_2_O_2_ [227]. Moreover, a diversity of PRDX isoforms can be found in most organisms within most cellular compartments [228,229,230,231]. For example, *E. coli* possesses three PRDX isoforms, *Saccharomyces cerevisiae* possesses five, and humans possess six [193]. The prevalence of multiple variants of the same protein hints at the extensive roles that PRDX fulfill, extending much beyond their antioxidant defense function [194].

Under conditions of acute oxidative stress, PRDX can undergo hyperoxidation, leading to various structural and functional changes in these proteins depending on the degree of oxidation (Figure 3). This modification leads to the inactivation of PRDX’s enzymatic activity, which can usually be reversed by the enzyme sulfiredoxin (SRXN1). This recycling reaction is slow enough to allow hyperoxidized forms to persist in the cell for many hours. Rhythmical oscillations in the abundance of hyperoxidized forms of PRDX show a circadian pattern (Figure 3) [191,202,232,233,234,235]. Importantly, although it inhibits peroxidase activity, hyperoxidation enables PRDX to perform cell signaling functions. Furthermore, under conditions of excessive oxidative stress, PRDX can adopt an irreversible hyperoxidized form that enables them to function as molecular chaperones, thereby reducing protein aggregation and increasing cell survival (Figure 3) [194,235,236]. The unique ability of PRDX to become hyperoxidized is hypothesized to be an evolutionary adaptation that links redox metabolism, timekeeping capabilities, and various other cellular functions of these proteins [178].

Bacterial PRDX are less sensitive to damage caused by overoxidation and thermal stress compared to their counterparts in more complex eukaryotes [235,237,238]. This is quite understandable, given that many microbes must withstand significantly high ROS levels generated by their hosts [239]. While the bacterial version is “stronger”, the eukaryotic one is much better adapted for signaling functions, acting as an assistant that facilitates cell communication [237,238,240].

It is well known that ROS can be very harmful to cells. The accumulation of ROS leads to oxidative stress, which results in a range of adverse effects, including impaired enzymatic activity, lipoproteins modifications, DNA and cell membrane damage, inflammation, exacerbation of misfolded proteins, and various other cellular detrimental effects [241]. Mice with a knockout of PRDX1 have a reduced lifespan of 15% [242]. However, despite being an oxidant, H_2_O_2_ is also involved in both intracellular and extracellular signaling [196,243,244,245].

To serve its signaling function while remaining harmless to cells, H_2_O_2_ requires tight regulation [246]. In comparison to PRDX, which are primary hypersensitive H_2_O_2_ receptors, other effector proteins involved in signaling react markedly slower with H_2_O_2_ [247]. Due to its high activation energy barrier [248], some authors bluntly describe the reactivity of H_2_O_2_ with other biomolecules as “sluggish” [249], at the same time emphasizing that this characteristic is not a defect, but rather a unique advantage of H_2_O_2_ compared to other highly reactive oxidants. This slow reactivity enables hydrogen peroxide to travel further away from its source, enabling it to reach more reactive targets at desirable locations [249].

The abundance and especially high reactivity of PRDX, which is several orders of magnitude higher than reaction rates displayed by other proteins [247,248], allow these proteins to react with the majority of H_2_O_2_ in human cells, up to 99% in the cytosol and over 90% in the mitochondria [250,251]. The ability to maintain strict control over H_2_O_2_ makes it an extremely useful molecule in contrast to free radical oxygen species, which mostly cause harm to biological molecules, contributing to the development of diseases [248]. H_2_O_2_ is a key component regulating a vast array of biological processes, including inflammation, cell proliferation and differentiation, immune responses, tissue repair, apoptosis, aging, and general metabolism [252,253,254]. Due to its exceptional properties like selective reactivity and relative stability, H_2_O_2_ is sometimes referred to as the most important redox signaling molecule [255].

To effectively perform their tasks, PRDX exhibit varying levels of activity at target sites [256]. PRDX molecules can become inactivated, which permits H_2_O_2_ to accumulate where it is specifically needed, thus allowing cellular processes to take place [193,257]. Concurrently, other PRDX remain fully operative, counteracting severe oxidative stress in the rest of the cell [226,236,237,258,259,260]. This hypothetical mechanism is referred to as the floodgate model [194,237,246,261].

PRDX can also act as sensor transducers of H_2_O_2_ through a mechanism known as a redox relay (Figure 3) [246,262,263]. In this case, PRDX mediate oxidation by initially receiving and then transferring oxidative equivalents to target proteins through thiol–disulfide exchange [260,264,265,266,267,268]. In this way, PRDX interact with a wide range of target proteins across various cellular compartments, modifying their structural conformation and functionality [258,269,270,271,272,273]. Without PRDX relay activity, certain target proteins, which likely possess interaction sites specifically for mediated oxidation [273], would not be able to be oxidized solely by H_2_O_2_. Hence, PRDX are necessary to maintain the proper functionality of redox-regulated target proteins and should not be seen as their competitors [266,268,273,274,275,276]. The redox relay mechanism provides an explanation as to why some intrinsically unreactive proteins can still be oxidized in the presence of antioxidant scavenging systems [267,277].

Intriguingly, under acute stress conditions, the floodgate and relay mechanisms seem to operate in opposite ways. In the redox relay mechanism, high concentrations of H_2_O_2_ result in PRDX hyperoxidation, which inhibits redox signaling through oxidation. This renders target proteins insensitive to oxidation or even causes them to become reduced, which has a protective effect and enhances cell survival [235,275,278,279,280]. Simultaneously, direct target protein-H_2_O_2_ interactions can occur, thus not excluding but giving the possibility of parallel operation of both mechanisms within temporally appropriate and specific cellular areas [194,260,273].

However, emerging perspectives indicate that there is only scant evidence supporting the floodgate model, and even in these rare cases, the observed effects may be opposite to those predicted [235]. In contrast, there is a growing body of evidence supporting the redox relay model, in which the pro-oxidative relay function of PRDX does not conflict with its antioxidant role [194,235,267,273,277].

It is noteworthy that various PRDX isoforms display distinct levels of sensitivity to hyperoxidation. However, for a long time, the molecular basis underlying these differences remained elusive. Recently, two novel hyperoxidation resistance motifs, termed A and B, were identified alongside the previously known YF and GGLG sensitivity-conferring motifs, which are typically absent in prokaryotic PRDX [239]. This discovery sheds light on the mechanisms by which PRDX fine-tune their sensitivity to hyperoxidation, assess whether H_2_O_2_ levels are hazardous, and explain how cells determine which signaling pathways should be activated in response to oxidative stress [239].

Moreover, the identification of these novel motifs elucidates the disparities in resistance levels between human, mitochondrial PRDX3, and cytoplasmic PRDX1 and PRDX2 [239] despite their high sequence similarity and the concurrent presence of YF and GGLG motifs [281]. Consequently, the A and B motifs constitute a conserved mechanism of PRDX resistance to hyperoxidation across both eukaryotic and prokaryotic organisms [200,239].

Interestingly, the significance of these novel motifs was validated using mutagenesis of *Salmonella enterica* PRDX. This Gram-negative bacterium is a predominant cause of bacterial foodborne-related illnesses and deaths [282], imposing a significant economic and public health burden [283,284,285]. Like *E. coli*, *S. enterica* is an enteric pathogen that competes with the gut microbiota, leading to gastrointestinal diseases and causing subsequent alterations in the genomic, taxonomic, and functional traits of the human microbiome. These interactions may facilitate the horizontal gene transfer that renders *Salmonella* more virulent and resistant to antibiotics [286]. It is conceivable that the improved comprehension of the mechanisms dictating the variable susceptibility of PRDX to hyperoxidation could shape future strategies in the treatment of infectious diseases. For example, enhancing the susceptibility of bacterial PRDX to inactivation could represent an innovative approach to the development of new antibiotics and pharmacotherapeutic interventions [239].

Furthermore, the authors of the study describing the discovery of novel motifs emphasize the potential application of PRDX-targeted pharmacological interventions in the field of oncology [239]. Most chemotherapeutic agents exert anticancer effects through the induction of oxidative stress, which perturbs the redox homeostasis of cancer cells and elevates intracellular ROS levels [287]. It may be possible to augment the susceptibility of cancer cells to oxidative damage by altering certain PRDX motifs such that PRDX becomes more sensitive to hyperoxidation [239]. Consequently, precisely targeted drug interventions aimed at modulating PRDX functions could potentially enhance the efficacy of a broad spectrum of chemotherapeutic treatments.

## 10. Mycobiome May Be a Window into the Intricacies of Biological Timekeeping

The human microbiota comprises not only a vast array of prokaryotic organisms but also a diverse community of eukaryotic species, including protists and fungi. The mycobiome, which refers to the fungal microbiome, is increasingly being recognized as a significant and integral component of the human microbiome. Mycobiome extensively interacts with gut bacteria, influences immunity, and is implicated in the exacerbation of several diseases [288,289,290,291].

In general, the mycobiome is dominated by yeasts, with the genus *Saccharomyces* being the most abundant [288,292]. Scientists remain uncertain as to whether these fungi constitute a “core gut mycobiome” or if their presence is simply a consequence of the ingestion of yeast-containing consumables [288,289,292,293]. Nonetheless, since colonization is not a prerequisite for commensal fungi to exert an impact on host physiology, the significance of their presence should not be underestimated [288].

*Saccharomyces cerevisiae*, commonly known as baker’s yeast, frequently serves as a eukaryotic model for biochemical and genetic studies, analogous to the use of *E. coli* as a model bacterium. Over the past 5000 years, *S. cerevisiae* has accompanied human evolution through food and beverage consumption [294]. It is commonly found in the healthy human microbiota of the oral cavity [295], intestinal tract [296,297,298], skin [299], and vagina [300,301]. The use of *S. cerevisiae*-based probiotic strains has been shown to give promising therapeutic outcomes in the management of Leśniowski-Crohn disease [302], irritable bowel syndrome, and ulcerative colitis and in the prevention of antibiotic-associated diarrhea, acute diarrhea [303], and bacterial vaginosis [301]. Probiotic yeasts can also cooperate with probiotic bacteria in many ways, potentially increasing their effectiveness [303]. However, although it has low virulence, *S. cerevisiae* can also become an opportunistic pathogen among individuals with compromised immune systems [304,305,306].

When it comes to its circadian properties, *S. cerevisiae* does not appear to possess any clock genes recognized in other fungi or animals. However, there is evidence for the presence of certain post-translational rhythmic markers that persist in the absence of transcription [307]. For example, baker’s yeast displays oscillations in metabolic activity, which share some mechanistic similarities with biological clocks. This potential timing mechanism is subject to circadian entrainment, demonstrating responsiveness to both the duration and intensity of environmental cues [43,307]. Specifically, *S. cerevisiae* exhibits fluctuations in pH levels, which show delayed phases relative to 24 h temperature cycles that mimic day–night changes. However, these oscillations have a weak free-running rhythm, which instantly damps in constant conditions [43,307].

Nevertheless, a phenomenon characterized by cell-autonomous, ultradian oscillations in oxygen consumption and metabolism known as the yeast metabolic cycle (YMC) has been found to exhibit robust rhythmicity during continuous growth under aerobic, nutrient-limited conditions [308,309,310,311,312]. Given that YMC shares a variety of features with biological clocks, it could be considered analogous to CC [313,314] and called an ultradian clock, as the periods of its oscillations range from dozens of minutes to several hours [310,312,315,316,317]. YMC is intrinsic, temperature-compensated, and can oscillate for months, also synchronizing to a variety of zeitgebers [221]. YMC facilitates the coordination of most cellular processes [313,318], additionally logically sequencing them [221]. Significantly, over 50% of the baker’s yeast genome is expressed cyclically via the YMC mechanism [310,319,320,321].

Importantly, there is a bidirectional relationship between the cell division and metabolic cycles, where the YMC gates essential cell cycle events, ensuring proper coordination between them [322,323,324,325]. YMC imposes metabolic checkpoints, which confirm that the cells have stored enough energy reserves to perform certain cell cycle stages [221,322]. This is crucial since yeast growth is overly sensitive to nutrient availability [326].

YMC oscillates between two metabolic stages characterized by varying levels of oxygen consumption. Processes such as autophagy and nutrient storage do not require significant oxygen consumption; hence, the period in which they occur is called LOC (low oxygen consumption), which occupies most of the duration of YMC [326,327]. When enough metabolic resources are gathered, cells enter the oxidative stage (HOC—high oxygen consumption), liberate carbohydrate stores, and enhance protein synthesis [326].

Initially, it was widely believed in the scientific community that YMC temporally separates cell division from oxidative metabolic reactions in such a way that crucial cell cycle events are restricted to the reducing phase [319,325,328]. The assumption was that critical cell division processes, such as DNA replication, do not occur during the HOC because this stage is correlated with high ROS levels. Shifting DNA synthesis to the transition period between HOC and LOC should potentially reduce the risk of oxidative stress-related damage and mutagenic consequences [216,308,310,319,325].

However, contrary to initial assumptions, subsequent years of research have revealed that DNA replication can occur in the early and middle stages of HOC, which can even synchronize exactly with the S phase of the cell division cycle [322,329]. As a result, the previously held hypothesis that DNA replication is entirely segregated from aerobic respiration has been challenged. Moreover, further investigations demonstrated that the relationship between YMC and the cell division cycle is very plastic [221,321,322,329]. Notably, it was established that DNA synthesis takes place towards the end of the HOC phase, which coincides with the peak of PRDX hyperoxidation, an indicator of oxidative damage [221,310]. Likewise, off-gas analyses show that changes in cellular oxygen consumption are directly inversely proportional to the concentration of dissolved oxygen [321,330,331,332,333].

The ambiguity in determining the reductive and oxidative stages of YMC has strengthened the arguments for the ROS-DNA synthesis hypothesis, which was based on flawed or misinterpreted results [332,334]. This inconsistency between predictions and results was not anything special. An inverse proportion between ROS and respiration intensity was already observed several decades ago, much earlier than the “erroneous” ROS-DNA hypothesis emerged [334]. Since the YMC does not consistently maintain a strict separation between DNA replication and HOC, it has been proposed that the cell cycle START checkpoint, specifically the G1/S transition [335], is coupled with the shift to HOC, which is marked by a notable rise in respiration [322].

Both YMC and CC are accompanied by changes in cellular redox state [221,313]. Moreover, it is known that individual, independently oscillating yeast microorganisms tend to synchronize with one another [308,336,337]. Almost ⅓ of cells in the yeast population undergo harmonized division during each YMC cycle, precisely coordinating pH and ROS concentrations, due to intercellular signaling [310,322,325].

Recently published, a remarkably interesting study confirmed that, indeed, fluctuations in cytosolic H_2_O_2_ production are opposite in phase to oxygen consumption rhythms [338]. Researchers discovered that after the deletion of the genes encoding both yeast’s PRDX Tsa1 and Tsa2, intercellular synchronization disappears, and a complete disconnection between metabolic cycling and cell division can be observed [338]. These results highlight that PRDX are essential for maintaining cell cycle synchrony and for coupling YMC with cell division, which also emphasizes the fact that metabolic cycles are independent of cell division [321,324,338].

Authors of the study have also noticed that the application of the reducing agent dithiothreitol can artificially extend HOC, which subsequently evokes the formation of double-budded cells. This finding shows that cells can pass START multiple times without finishing the division cycle correctly. On the other hand, the oxidizer diamide prevents the progression from the G1 phase, decreases DNA replication, and impairs growth [338]. It appears that the transition to HOC is essential for the initiation of the cell cycle START and passing through the G1/S checkpoint, whereas the completion of cell division necessitates a transition to LOC, which, when prolonged, seems to “trap” cells in the G1 phase [338].

Another emerging finding from this study suggests that PRDX are involved in regulating YMC cycles through the redox relay mechanism. Unlike the floodgate model, the redox relay can explain why LOC-HOC switching was observed after adding sufficiently substantial amounts of H_2_O_2_. The previously discussed process, upon which induced PRDX hyperoxidation results in target protein reduction, accurately matches the obtained results [338].

Of note, ultradian oscillations in gene expression, redox state, and metabolism have also been detected in protists and two other yeast species, *Candida utilis* [339,340] and *Schizosaccharomyces pombe* [339,340,341,342,343]. Investigations on *S. pombe* have strengthened the view of the redox relay mechanism and the presumed result of PRDX hyperoxidation on target protein reduction, opposing the assumptions of the floodgate model [274,280]. In this context, hyperoxidation enhances PRDX’s chaperone function and allows for the accumulation of reduced thioredoxin, which in turn can focus on reducing proteins other than PRDX (Figure 3) [196,236,279,344].

Clinical strains of *S. cerevisiae*, when incubated in human blood, exhibit significantly higher levels of Tsa1 expression than their avirulent counterparts. This heightened oxidative stress response allows them to withstand oxidative burst attacks from human neutrophils and macrophages much better than non-clinical strains, like pathogenic *Candida albicans* [345]. Thus, under exposure to high concentrations of H_2_O_2_, virulent strains exhibit a higher survival rate. Accordingly, in another life-threatening opportunistic fungal pathogen, *Candida glabrata*, Tsa1, Tsa2, and SRXN1 have been recognized as crucial components for its survival in a pro-oxidative environment promoted by the host immune system [346].

In addition, *S. cerevisiae* cells deficient in all PRDX variants cannot alter the gene expression of nearly 50% of transcripts in response to increased ROS levels, thereby losing their ability to transcriptionally respond to H_2_O_2_ signals [267]. Theoretically, PRDX may regulate various transcriptional and metabolic processes through redox modulation of the AMP-controlled protein kinase A (PKA), which in turn targets other stress-responsive transcription factors. Therefore, the anti-aging effects of PRDX activity should be attributed more to their signaling and chaperone functions and, to a lesser extent, their H_2_O_2_ scavenging function [44,344,347,348,349]. In baker’s yeast, the prevalence of PRDX isoforms is remarkably high, comprising up to 1% of the total soluble proteins in the cell, significantly outnumbering both GPX (50-fold) and catalases (500-fold) [350]. The presence of high concentrations of PRDX inhibitor causes yeast cells to die within 24 h [313].

Collectively, these data underscore the importance of PRDX in yeast as vital antioxidant enzymes capable of exhibiting other critical cellular functions while also suggesting their possible evolutionary origins alongside circadian systems, especially given that the oxidation phase of PRDX is linked to yeast’s respiratory cycle.

Importantly, the connection between the CC and cell division is not exclusive to the YMC and has been observed in diverse organisms spanning from bacteria to humans [351,352]. Exploring the intricate relationship between the cell cycle, PRDX, and CR in cancer cells could lead to significant advancements in cancer therapy [314,338,352,353].

## 11. The Crossroad of Metabolic and Antioxidant Functions within the Timekeeping Framework

Ultradian clocks have been proposed as potential archetypes of CCs [354,355,356]. Their relevance is increasingly recognized from unicellular eukaryotes to humans, with the redox cycle being the centerpiece of their mechanisms [312,318]. On the other hand, it was also suggested that YMC and other ultradian rhythms could primarily have originated as diel cycles, but their length decreased due to evolutionary factors [221,311]. Regardless of the solution to this issue, what constitutes the core of timekeeping mechanisms and the primary evolutionary reasons behind their current form remain enigmatic.

Unlike many other eukaryotic organisms, *S. cerevisiae* lacks well-characterized photoreceptors and was previously thought to be unreactive to visible light [212,357]. However, a dramatic shift in the phase of YMC oscillations in response to H_2_O_2_ has been identified as analogous to the light-induced entrainment seen in the phase-response curve of the classical circadian system [325]. Moreover, it has been observed that even direct exposure to natural sunlight through a window can alter the ultradian metabolic rhythm of baker’s yeast [212].

Yeast cells use a peroxisomal oxidase to convert light into an H_2_O_2_ signal that is subsequently detected by Tsa1 and relayed to thioredoxin, which inhibits PKA activity and helps regulate the accumulation of Msn2 in the nucleus [44]. Msn2 is a PKA-regulated transcription factor critical for triggering the expression of stress-responsive genes upon exposure to various environmental signals, including oxidative stress [358,359]. Its deletion disturbs LOC to HOC switching, leading to YMC dysregulation [338,360]. This potentially explains how yeast cells can sense light without photoreceptors, highlighting the importance of the H_2_O_2_ signaling function. These findings align with other research, which observed that in the case of zebrafish, H_2_O_2_ can act as a second messenger that conveys photic signals to its circadian system [361].

PRDX are essential for yeast’s robust and stable ultradian metabolic function. Hydrogen peroxide, starring as a second messenger, offers a connection between light-sensing, internal physiology, and timekeeping [44]. The rhythmic nature of ROS generation by underlying oxidative metabolism is very likely to be responsible for PRDX redox oscillations, which at the same time seem to be a cellular clock’s input and output [219,221,313,362]. However, a question arises: why is there such a strong connection between the antioxidant, timekeeping, and signaling systems and cellular metabolism?

Although the traditional view suggests that the first organisms were strict anaerobes, new findings indicate that LUCA could tolerate minimal levels of oxygen. Phylogenetic analyses demonstrate that LUCA may have possessed metabolic pathways for ROS detoxification and even those involving oxygen and H_2_O_2_ utilization (or at least certain components thereof), which could lay evolutionary foundations for aerobic respiration [363].

This alternative scenario is increasingly gaining credibility, especially given that H_2_O_2_ was present during the emergence and evolution of life on early Earth, acting as one of the earliest oxidants and an important source of molecular oxygen. H_2_O_2_ and O_2_ can be abiotically produced through various photochemical and nonphotochemical processes, including interactions between sunlight and atmospheric components, as well as reactions at mineral-water interfaces (Figure 4, point 2) [363,364,365,366,367]. Therefore, initial ROS control systems probably originated in response to slightly oxic microenvironments localized within the anoxic surroundings of early Earth.

Accordingly, the presence of low yet notable levels of ROS led to the evolution of PRDX approximately 3.8 billion years ago, twice as early as the Great Oxygenation Event and well before the evolution of the first cyanobacteria [368]. The evolution of primitive ROS control pathways facilitated the use of ROS as the earliest signal transduction molecules. This advancement empowered primeval organisms with intra- and inter-cellular communication capabilities while also protecting the first cyanobacteria from the harmful effects of oxidative stress. In this context, the emergence of primordial antioxidant protective systems allowed for the utilization of H_2_O_2_ as an electron donor for early photosynthesis. This suggests that even a minimal ROS supply was crucial for the development of oxygenic photosynthesis from anaerobic photosynthesis [365,368].

Sunlight, regarded as the inaugural zeitgeber, shaped early circadian systems well before acute oxidative conditions arose (Figure 4, point 1) [358]. Given that even the most primitive forms of life had to cope with locally low but consequential aerobic conditions (Figure 4, point 2), metabolic and redox pathways must have naturally co-evolved with ancient timekeeping mechanisms (Figure 5). In eukaryotes, the primary purpose of biological oscillations, such as YMC or CR, is speculated to optimize metabolic resource utilization while minimizing protein homeostasis costs [326].

Energy supply and oxidative stress might be the most important environmental factors that have been driving the evolution of molecular anticipation mechanisms [202]. Therefore, it is very convincing that metabolic oscillators interlinked with antioxidant systems represent the ancient timing mechanisms, which served as a rudimentary foundation upon which the variety of ultradian and circadian biological clocks of present-day organisms have been built [56,179,309,311,312,313,318,338,355,369,370,371,372,373].

Almost all of the currently understood classical circadian systems lack a common underlying principle of operation, whereas metabolic rhythms appear to be conserved across all organisms [202]. With the course of evolution, primordial timekeeping mechanisms have ultimately evolved into the currently known form of circadian systems comprising both PTO and TTFL mechanisms (Figure 5).

Regarding the co-evolution of metabolic and redox pathways with ancient timing mechanisms, over billions of years of advancement, primordial timekeeping mechanisms have ultimately evolved into the diversity of circadian systems seen at present. Many factors have influenced the development of biological clocks in various species. The maintenance of a specific form of a biological clock in a particular species depended on the intensity of the interaction with a given factor. The primary driving force behind the evolution of biological clocks has been the influence of cyclic, circadian stimuli, which, through circadian entrainment, led to the adaptation of metabolic and redox pathways for the purpose of timekeeping. These rhythmic pathways were subsequently gradually enhanced with additional timekeeping mechanisms. As a result, the currently observed biological clocks have emerged, characterized by the coordinated operation of the transcription–translation feedback loop (TTFL) and post-translational oscillation (PTO) mechanisms.

PTO can operate independently from TTFL, enabling CRs to persist in the absence of transcription [190,191,202,374]. However, this does not mean that they operate separately. Rather, there is a bidirectional link between PTO and TTFL mechanisms. PTO and cytosolic processes can relay timing information to the transcriptional clock [56,178,370,375,376,377], and TTFL can influence metabolic and redox cycles by altering metabolite production [56,311,370,378,379].

In zebrafish fibroblasts, acute hypoxia alters oscillations of cellular metabolism and intracellular H_2_O_2_ distribution and induces a rapid shift in the redox state. Eventually, the readjustment of metabolic redox rhythms results in TTFL attenuation. In the absence of transcriptional mechanisms, NADH, NADPH, and PRDX redox systems ensure the maintenance of the cell’s autonomous rhythm of cytosolic H_2_O_2_ [380]. In the case of the smallest free-living unicellular eukaryote, *Ostreococcus tauri*, which has a light-entrainable CC, constant darkness does not terminate its CR despite the cessation of transcription and translation processes [191].

Consequently, the eukaryotic CC model should be understood as a product of the interplay between transcriptional circuits and non-transcriptional factors [56,185,370]. The cellular clock’s interaction with redox signaling is dependent on both transcriptional and non-transcriptional mechanisms, which could be viewed as cogs of one, bigger cellular clockwork since both are essential for the robustness of CRs [370,381]. It is plausible that diverse transcriptional clocks have evolved to serve species-specific functions in response to distinct environmental niches, which is supported by the fact that TTFL evolved independently multiple times during evolution [56,178,182,356]. Thus, TTFL could be perceived as a time-telling mechanism that operates based on the underlying layer of metabolic activity [56,382].

The most ancient components of biological clocks are non-transcriptional in nature, predating the existence of transcriptional rhythms [191,192,202]. Importantly, the inactivation of PRDX or SRXN1 only perturbs but does not terminate YMC and circadian oscillations [313,383,384]. Cycles of hyperoxidized PRDX are dependent on proteasome degradation, and they vanish due to proteasome inhibition [383,385]. Therefore, it is tempting to suggest that the primary, fundamental time-keeping mechanism is not encoded in PRDX oscillations but, in the light of the latest research, determined by the putative metabolic rhythms that they report on [56,178,202,338,382].

## 12. Metabolic Rhythms: The Heartbeat of Circadian Timing

The pentose phosphate pathway (PPP) is a vital metabolic pathway that functions in parallel to glycolysis. Its primary role involves the generation of the redox cofactor NADPH, a critical reducing agent required for a multitude of biosynthetic reactions and antioxidant defense mechanisms within cells [386]. As demonstrated in human cells, mouse tissues, and living flies, altering the PPP can influence CR, affect clock gene expression, and have a profound impact on circadian redox oscillations, as NADPH rhythms are a prerequisite for PRDX hyperoxidation cycles [387]. YMC, with its circadian properties, is also highly dependent on NADPH abundance [388]. Moreover, in zebrafish fibroblasts, the PPP seems to act as a crucial regulator of circadian timekeeping. The enhancement of metabolic rhythmicity in cooperation with the redox system preserves zebrafish cells’ CR under hypoxic conditions when TTFL is not operational [380].

Furthermore, recent research on human red blood cells confirmed that circadian oscillations in the redox state are tied to the core rhythmic glucose metabolism [389]. The rhythmic regulation of glycolytic and PPP fluxes displays opposite phases. The circadian night is correlated with the peak in glycolytic flux, whereas the circadian day is associated with the peak in PPP flux, which also aligns with the maximum PRDX oxidation. Within this framework, ROS generation acts as a zeitgeber, entraining the cellular redox state with the involvement of NADPH. The redox balance can thus communicate with core metabolism and synchronize observed rhythms with the host’s physiology [389]. This research elegantly explains how glucose metabolism and redox factors collaboratively function to generate CR in erythrocytes, cells incapable of gene expression, which also do not possess any organelles but still exhibit rhythmicity. This rhythmic behavior was also confirmed in mouse fibroblasts devoid of Bmal1, a fundamental component of the mammalian CC [389].

Additionally, in mouse embryonic undifferentiated stem cells, circadian oscillations in glucose utilization manifest before the expression of canonical circadian genes [390]. In drosophila S2 nucleated cells, which lack the key circadian genes that could form TTFL, there are still temperature-compensated circadian metabolic oscillations in central carbon metabolism and amino acid metabolism coupled to gene and protein cycles [391]. Hence, even in eukaryotic systems, cellular metabolism is an essential part of the PTO mechanism [389].

Regarding cyanobacteria, studies also indicate a close interrelation between their metabolism and CC, where the latter fundamentally represents a PTO mechanism embedded within a TTFL framework [61,392]. The addition of glucose can impact the expression of *kai* genes and induce a delay in CR [393]. Moreover, the cyanobacterial CC is not directly light-sensitive but instead responds to the cell’s metabolic state, including the redox state of the plastoquinone pool [394] and the ATP/ADP ratio [395]. Both these factors reflect photosynthesis activity and are essential for entraining the cyanobacterial CC. Based on this, it could be assumed that a sort of “metabolic feedback loop” is formed. Metabolic inputs induce fluctuations in the energy storage metabolism, which, by acting as an output, make metabolic rhythms that eventually close the loop [396]. Furthermore, PPP and glycogen metabolism are crucial for initiating photosynthesis during the transition from darkness to light [397]. Even without any light–dark cues, cyclical glucose feeding induces ATP/ADP ratio fluctuations, demonstrating that the cyanobacterial CC can operate as a generalized metabolic timing system. Therefore, irrespective of their source, it is plausible that metabolic oscillations may be the primary driver of the *kai* system [398]. These data underscore how tight the bidirectional connection is between cyanobacterial CC and metabolic rhythms, which, by working in tandem, ensure the precise synchronization of cell physiology with environmental cues [64]. The possibility that the well-defined CR observed in *B. subtilis* arises from metabolic mechanisms is also under consideration [32,80].

In conclusion, the growing body of evidence increasingly supports the hypothesis that primeval and fundamental biological timekeeping mechanisms might be rooted in metabolic oscillators characterized by the interplay and periodic activity of highly conserved central metabolic pathways, along with their principal elements and components [56,178,202,221,355,371,387,389,391,399].

However, many arguments oppose this conclusion, signifying that much remains unknown [400]. Redox and metabolic rhythms could merely be the inputs and outputs of an as-yet-unidentified biological clock rather than its crucial components. In mammals, for instance, interrupting the activity of energy or redox metabolism does not change the circadian period, implying that neither of these pathways constitutes an essential timekeeping mechanism [219,400]. However, this is primarily based on higher eukaryotic models, in which the TTFL mechanism has surpassed PTO in advancement during the evolution [49,202].

In baker’s yeast, primary metabolism exhibits oscillations that provide suitable conditions for the temporally varying demands of biosynthetic processes. Hence, the genesis of metabolic oscillations may not necessarily originate from the primary metabolic pathways themselves. They might instead result from the activity of the putative biosynthetic oscillator. Its mechanism could be constructed on the foundation of negative feedback between various biosynthetic processes, potentially anchored in the competition for metabolic resources or signaling pathways, such as the Target of Rapamycin (TOR) pathway [333]. Nevertheless, this does not mean that the two perspectives are entirely incompatible. It is conceivable that both metabolic and biosynthetic oscillations interact in complex ways, eventually contributing to the overall temporal organization and timekeeping within cells. Further research would be required to unravel these mechanisms completely.

In the context of potential eukaryotic metabolic oscillators, the TOR pathway emerges as a promising avenue for investigation. TOR is a highly conserved protein kinase pathway involved in nutritional sensing and growth regulation [400]. Interestingly, TOR was not only recognized as CC’s output pathway in several eukaryotic organisms but it has also been found to exhibit and sustain rhythmic activity independently of the traditional TTFL. In the case of the fungus *Neurospora crassa*, in which the CR cannot be attributed to the TTFL mechanism alone, TOR is strongly involved in the functioning of its CC. It was suggested that the TOR signaling pathway and its regulatory mechanisms might form negative feedback loops with other metabolic pathways or cellular events. Collectively, these could potentially function as quasi-universal metabolic oscillators across eukaryotic organisms. However, considering that red blood cells do not possess the TOR pathway, yet their central metabolic pathways demonstrate robust rhythmicity, various metabolic oscillators may exist inside cells [400,401,402].

According to a model of CC evolution proposed by Roenneber and Merrow, the synchronization of multiple overlapping rhythmic metabolic feedback loops, even of ultradian nature, could eventually give rise to a self-sustaining circadian oscillator [355,403].

Since its origin, cellular metabolism has been influenced by pronounced daily environmental cycles, and first life forms were certainly equipped with the ability to form complex networks of feedback loops. Most likely, both LUCA and LECA (Last Eukaryotic Common Ancestor) possessed many metabolic and redox pathways [363]. Therefore, the emergence of early timing mechanisms based on interconnected metabolic cycles might have been a straightforward evolutionary transition (Figure 5) [355,403]. When functioning in synergy within a network, these loops could eventually form an initial, ancestral timing mechanism.

In this context, a central oscillator could be built with merely three to five coupled negative feedback loops of non-circadian oscillators that dampen under constant conditions. The consolidation and orchestration of the initially chaotic responses from this primitive mechanism could be achieved by employing the clock’s output as one of its inputs. From this point, the development of an endogenous and autonomous circadian oscillator would necessitate only a few additional evolutionary steps [355,356,404].

## 13. Paradigms Change around the Clock

It is interesting to observe how the scientific perspective on the evolution of CRs has changed significantly over the past few decades. Had this article been written in the early 1980s, eukaryotes would have been the primary focus when discussing the origins of the first timekeeping mechanisms. At that time, prokaryotes were simply believed to lack the necessary cellular complexity required for the formation of CR, particularly given that TTFL needs a nucleus for transcription [59,61,405]. Also, the belief that the lifespan of a single cell is much shorter than 24 h was a strong counterargument for such considerations. These beliefs set a strong bias among chronobiologists, discouraging them from exploring CR in prokaryotes [60].

The discovery of CR in cyanobacteria in the late 1980s [65,406,407] marked a pivotal moment that irretrievably changed the view of the scientific community on cellular clockwork. Paradoxically, this advancement brought the science of CR to a dead end. The TTFL model became a prevailing paradigm, with researchers expected to conform and provide a rationale for it, even in the face of other possibilities [178,408]. Currently, it is recognized that in most organisms, cellular CRs result from the interplay between genetic regulatory oscillations and post-translational processes (Figure 5).

Why do cells possess numerous potential backup or alternative timing mechanisms? The presence of such redundancies suggests that CRs are critically important for the optimal functioning of biological systems. These mechanisms can offer additional safeguards against disturbances to the timing of biological processes when cells are influenced by internal or external factors [356]. In cyanobacteria, it is evident that the coupling between PTO and TTFL systems assures the robustness of the operation of the CC. A biological clock based on two mechanisms can continue to operate effectively enough in the situation when one of them fails [185,409,410,411].

However, in eukaryotes, the TTFL appears to be a more global mechanism of action. PRDX oscillations seem to be critical for maintaining cellular redox balance, serving signaling function, and ensuring reliable clock operation by fine-tuning the timing of the clock. Although the oscillations in redox cycles are not necessary for the circadian system to function, they play an important role in maintaining the accuracy and strength of the system.

Cellular timekeeping did not evolve directly into well-defined and robust circadian systems that exhibit self-sustained rhythmicity. The earliest organisms were evolving in the cyclic nature of Earth, where stable environments were rare. It is perhaps worth considering that evolution prioritized the development of the entrainment mechanism and not the self-sustained oscillator itself. Therefore, placing too much emphasis on discovering putative circadian mechanisms by examining them in constant conditions may underestimate the usefulness of damped oscillators [43,307,412]. Sufficient anticipation of environmental changes could be achieved even with rudimentary hourglass timers. A molecularly complex self-sustaining oscillator might simply represent the most optimal functioning state of an entrained damped oscillator, which was evolutionarily refined into an autonomous clock [164,355,412].

Specifically, with evolution, damped oscillators might have gradually evolved into more robust timekeeping systems by strengthening the coupling between metabolic components of the network and introducing additional regulatory components. This could potentially explain the diversity of CCs observable today (Figure 5) [355,356,412]. Primitive metabolic oscillations facilitated the efficient use of energy resources in nutrient-scarce conditions of early environments. Over time, this metabolic clock was “updated” with genetic mechanisms to fit the unique physiology of certain species [56,221]. CRs, hibernation behavior, and even sleep–wake cycles could potentially all be viewed as indications of an intrinsic interplay of metabolic cycles [371].

The intricately complex circadian systems of higher organisms, predominantly based on the TTFL mechanism, are a manifestation of evolutionary pressures, which shaped such complexity to ensure that biological timekeeping matches the sophisticated behaviors and structures of continuously evolving species (Figure 5) [49]. It is also possible that by incorporating additional feedback loops, evolution might have reinforced timekeeping systems against external noise, leading to the emergence of self-sustained oscillators [356].

Which of the three mechanisms—hourglass timer, damped oscillator, or sustained oscillator—will be favored and maintained in an organism largely hinges on the environmental conditions in which a given species thrives? Under relatively stable conditions, the most basic hourglass timer may be sufficient, whereas, in more unpredictable environments, robust systems with characteristics closer to well-defined CC are likely to be more advantageous [164]. Furthermore, the cyanobacteria of the genus *Prochlorococcus*, which is potentially the most prevalent photosynthetic organism on Earth, may have “de-evolved” its KaiABC complex, losing the KaiA component, yet remaining efficient in the stable environment that this bacterium resides [164,413].

Thus, is observing a strictly defined CR in organisms truly crucial? Nature is not concerned about definitions but rather prioritizes existence. Even the simplest mechanisms, such as an hourglass timer, can efficiently enhance survival chances. Irrespective of whether the timing system is self-sustaining or not, it functions similarly within a diurnally cyclic environment, providing adaptive advantages to specific organisms [165,171,356]. Individual organisms possess many various timing systems that surface only under conditions [356]. Emphasizing the entrainment mechanism within experimental approaches might reveal numerous unknown timing systems, as in the case of *S. cerevisiae* and *Caenorhabditis elegans* [43]. In the case of *K. aerogenes*, the bacterium may produce rhythmic patterns solely relying on signals from its host [82].

## 14. Time Cues That Facilitate Rhythmic Cooperation

Recently, a groundbreaking study demonstrated the maintenance of transcriptional rhythmicity despite the absence of a functional TTFL in Bmal1-deficient mouse cells [414]. However, this finding has sparked significant controversy among chronobiologists, leading to an intense debate regarding the validity of these results [415,416,417,418]. Whether eukaryotic redox and metabolic rhythms act solely as inputs and outputs or constitute the core of eukaryotic cellular timekeeping remains to be elucidated. Nonetheless, it is possible that in prokaryotic species, which make up most of the microbiota population, some kind of metabolic oscillator may serve both time-keeping and time-telling functions, concurrently implementing a temporal strategy enhancing the efficiency of the antioxidant defense system, as exemplified by the “basic” system seen in erythrocytes [178,389].

In a healthy and balanced gut microbiome, obligate anaerobes constitute most bacteria. This is due to the regulatory mechanisms exhibited by both the host and microbes, which, already from the moment of birth, create a hypoxic condition, favoring species adapted to low-oxygen environments. Therefore, elevations in oxygen levels or the presence of inflammatory conditions pose a threat to the gut microbial community, leading to intestinal dysbiosis [419,420,421]. The host can exert additional control over the location of the microbiota to maintain gut homeostasis and prevent unnecessary inflammation initiated by the immune response of the intestinal epithelium. This is achieved through the production of luminal H_2_O_2_ by the enzyme NADPH oxidase 1 (NOX1) in gut lining cells. NOX1-derived H_2_O_2_ restricts microbial growth and preserves a buffer between the microbes and the colon surface [420,422].

Interestingly, a recent study observed significant growth retardation in Bmal1-deficient cynomolgus monkeys, which has been suggested to be caused by spatial and temporal microbial dysbiosis [423]. The disruption in the normal composition and fluctuations of the gut microbiota was attributed to disturbances in the rhythmic release of luminal H_2_O_2_. Bmal1 regulates the transcription of intestinal NOX1, which in turn produces H_2_O_2_, thereby maintaining the rhythmicity of the gut microbiota. These results highlight a potential mechanism through which the host’s CC might influence the rhythmic oscillation of the gut microbiome [423].

Could H_2_O_2_ be considered a zeitgeber for timing mechanisms for at least some microbiota species? Research suggests that H_2_O_2_ may indeed serve as a circadian time cue both for eukaryotes and prokaryotes [44,361,424,425]. The prospect that many microbiota species may possess putative metabolic oscillators interconnected with antioxidant systems, along with the fact that H_2_O_2_ has been crucial in shaping timing systems since the beginning of life on Earth, further supports this exciting assumption. 

Gut microbes, although they do not have to directly perceive light, still must cope with the rhythmic environment of their host [10]. The rhythmicity of the microbiome living in low-oxygen conditions depends on many factors, which are also strongly influenced by the host’s CC. Among these factors, the influx of hormones (especially melatonin), changes in the host’s body temperature, eating habits (including meal timing), and hydrogen peroxide appear to serve as time cues for microbial timing systems. Metabolites produced by the microbiome also function as information that alters the host’s physiology, thereby affecting its CC (Figure 6).

Regular eating patterns of most animals give rise to daily changes in the intestinal environment, which in turn affect the gut microbiome [164]. Hence, it is no wonder that rhythmical food consumption, thus oscillations in nutrient availability, seems to be a crucial time cue for microbiome rhythms [11,12]. Mice with deleted circadian genes *Per1/2* recover previously lost microbial oscillations when subjected to scheduled feeding [21].

Furthermore, the human body temperature undergoes natural circadian fluctuations, which depend on several factors, including physical activity, sleep, and food intake. These temperature changes impact the composition and function of the gut microbiome [426,427]. Conversely, commensal gut bacteria may affect their host’s metabolism, thermogenesis, and thus body temperature. Some authors even suggest that the microbiome could be a plausible factor contributing to the progressive decline in average human body temperature observed over the last hundred years [426,428,429].

The microbiome can also experience daily variations in hormones secreted rhythmically by the host, such as melatonin [430]. Surprisingly, the gastrointestinal tract is the main source of human melatonin, where gut cells surpass the pineal gland in producing this hormone by 400 times. The secretion of melatonin appears to correspond with the frequency of eating [431]. Melatonin influences and regulates the circadian rhythms of the gut microbiota. Moreover, in the case of microbiome disturbance, exogenous melatonin application can recover microbial oscillations, as observed in mice fed with a high-fat diet [432].

Certainly, the ability to predict the arrival of fresh nutrients, certain hormones, or circadian body temperature changes could be very advantageous for microorganisms living inside the gastrointestinal tract [12,25,164]. Looking at how rhythmic the human body is, it would be almost illogical from an evolutionary point of view for at least a portion of the gut bacteria population not to have any internal timing mechanisms. These possessing biological clocks could control and fine-tune the activity of the rest of the symbiotic microbes, facilitating microbiota rhythms [25]. As exemplified by recent discoveries, the timekeeping systems of at least two gut microbial species may take advantage of host-provided zeitgebers [32,33,82]. Moreover, any kind of timing mechanism would be particularly useful for microbes after leaving the host, enhancing their viability in the external environment [164].

## 15. Concluding Remarks

The significance of biological timekeeping is evident throughout all living organisms, ranging from prokaryotes to humans. However, the remarkable complexity and diversity of mechanisms among known oscillators make them challenging to fully comprehend, especially at a fundamental level. Moreover, given that the beginnings of biological clocks might be traced back to the origins of life on Earth, uncovering the ancient timing mechanism proves even more intricate.

This narrative review covers the diversity of structures and functions of biological clocks that, reaching beyond the strict definition of a circadian clock, reveal how extensive the nature of biological timekeeping is. The discovery of timing mechanisms in non-photosynthetic bacteria thriving in such challenging, light-deprived environments as the gut only emphasizes how crucial the ability to track time is for life. By presenting how scientific understanding has historically “expanded” the presence of biological clocks across a wide array of known life forms and by resolving uncertainties regarding the function and activity of the yeast metabolic cycle or peroxiredoxins, the path that chronobiologists had to undertake to challenge subsequent dogmas was presented. This article serves as a comprehensive overview, gathering, organizing, and summarizing the available information and evidence concerning the evolutionary history of biological clocks.

Certainly, factors such as variations in light intensity, the availability of metabolic resources, and oxidative stress have collectively put evolutionary pressure on the convergent formation of the components of metabolic pathways, timing mechanisms, cellular communication, and antioxidant systems across ages. Many hypotheses about the source of cellular timekeeping have been proposed, but just as many have been disproven. Of the current theories, metabolic oscillators stand out as a promising area of study. The role of metabolic rhythms in biological clockwork is drawing increased attention, with emerging findings continually reinforcing their pivotal importance.

Biological clocks and metabolism have a bidirectional relationship, where they not only impact each other but also receive feedback from one another. In like manner, it is the symbiotic relationship between humans and their microbiome. The rhythmic cooperation between species of the metaorganism is undoubtedly an evolutionary masterpiece. A better understanding of how microbial biological clocks operate could shed light on how they contribute to maintaining health or how they influence disease progression, offering invaluable potential to revolutionize medicine. However, the core of the timing system still remains enigmatic, and there are still many questions that must be answered.

## Figures and Tables

**Figure 1 ijms-24-16169-f001:**
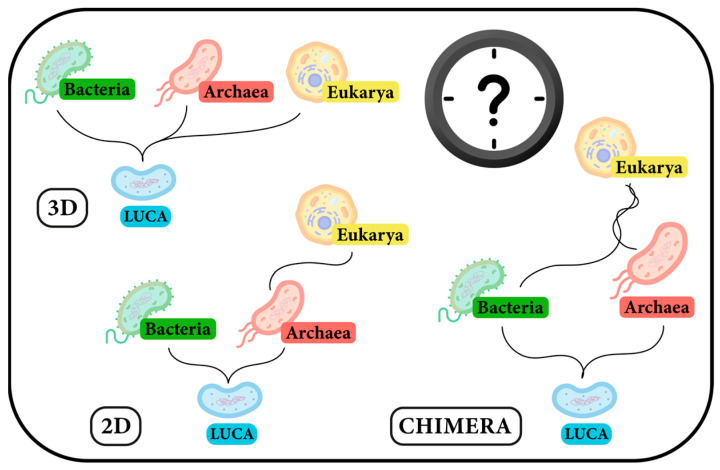
Visual representation of three different models describing the origins of Eukarya. CHIMERA—eukaryotes emerging as an evolutionary combination of bacterial and archaeal characteristics.

**Figure 2 ijms-24-16169-f002:**
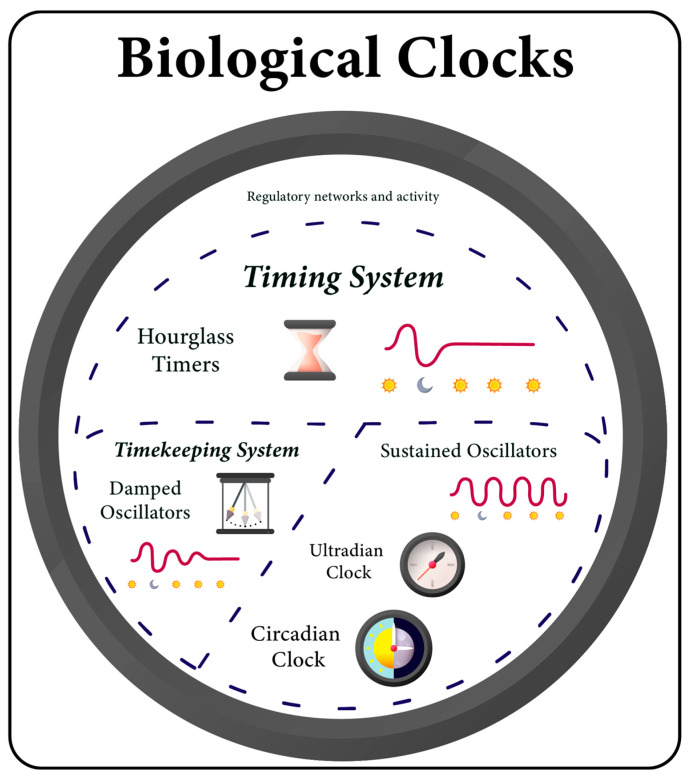
Overview of the different types of biological clocks in living organisms.

**Figure 3 ijms-24-16169-f003:**
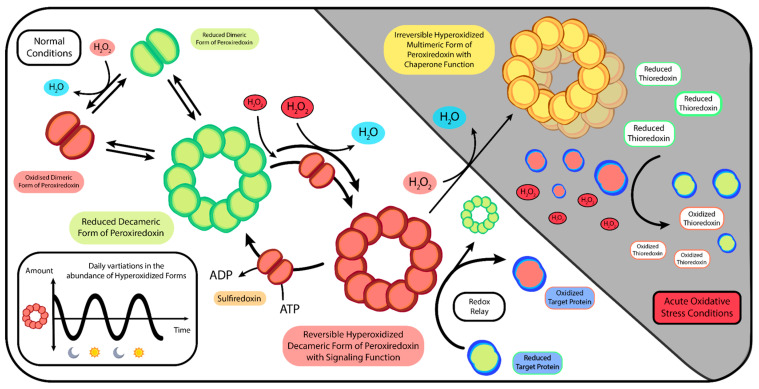
Visual representation of the diversity of structures, functionalities, and activities of peroxiredoxins under normal and acute oxidative stress conditions. A generalized model of redox relay signaling and the circadian cyclical variation in the abundance of peroxiredoxins is illustrated.

**Figure 4 ijms-24-16169-f004:**
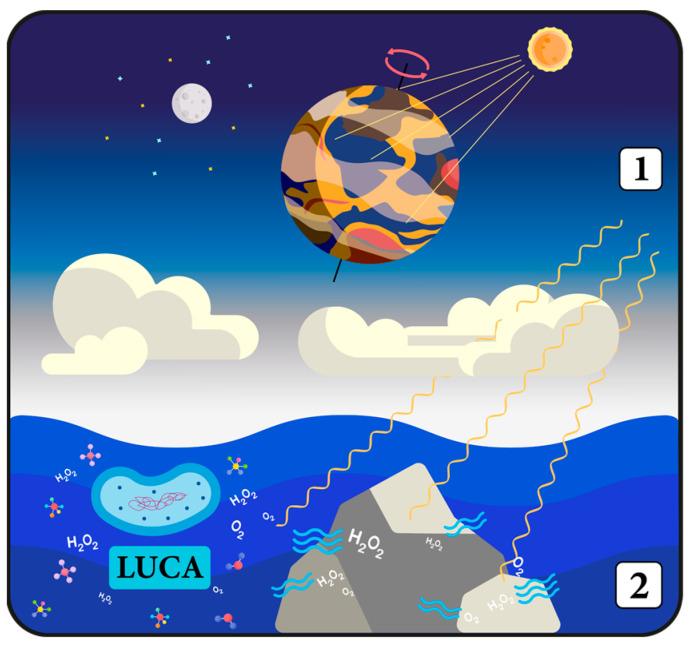
Key factors influencing the evolution of biological clocks: 1. The evolution of early Earth was shaped by constantly alternating periods of light and dark. Sunlight acted as the initial “zeitgeber”. 2. Primordial life forms had to cope and adapt to fluctuating nutrient availability and locally low aerobic conditions. Minimal (but significant) levels of oxidants were created abiotically through various photochemical and nonphotochemical processes.

**Figure 5 ijms-24-16169-f005:**
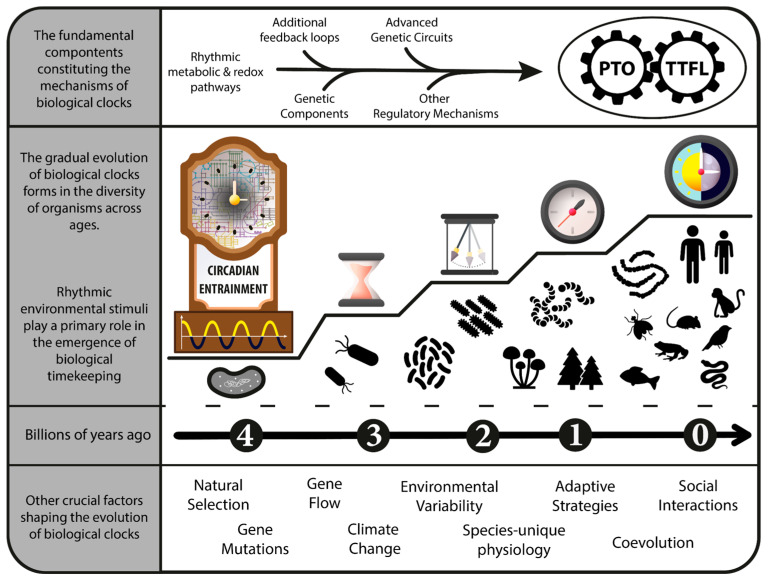
Key factors influencing the evolution of biological clocks.

**Figure 6 ijms-24-16169-f006:**
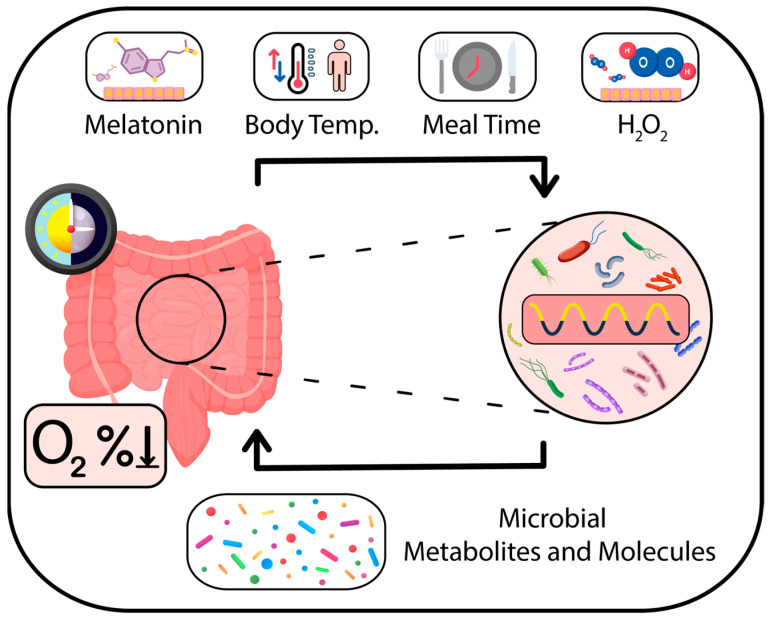
The bidirectional relationship between the microbiota and the host.

## Data Availability

Not applicable.

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
