# Peer review of "Studying the Human Microbiota: Advances in Understanding the Fundamentals, Origin, and Evolution of Biological Timekeeping"

_ijms, 2023, doi:10.3390/ijms242216169_

Round 1
Reviewer 1 Report
Comments and Suggestions for Authors
The review article (Studying the human microbiota: advances in understanding the fundamentals, origin, and evolution of biological timekeeping) by Adam and co delves into the evolution of the biological clock while focusing on the human microbiota, providing valuable insights into its fundamentals, origin, and advancements in our understanding of biological timekeeping.
Overall, the review is well written, and the literature cited is UpToDate however, there are still many minor issues that need to be addressed.
1. In the abstract, line 14, the term "anucleate" appears to be inappropriate in this context. It should be replaced with "prokaryote."
2. In the introduction (lines 60-64), it is stated that "well emerging evidence indicates that at least some microbiota…". It is advisable to include additional references to substantiate this claim.
3. Lines 282-284 mention "Combining time-of-day specific antibiotic administration…". It's important to note that Francesca et al. did not discuss day or night-specific antibiotics, and the concept of a circadian clock's relevance to antibiotic therapy in the context of pathogenic bacteria is unclear. Since most antibiotic resistance arises from DNA mutations rather than the regulation of specific gene expression in line with circadian rhythms, this point requires clarification.
4. Subheading 12 (lines 1031-1138) titled "Metabolic rhythms: the heartbeat of circadian timing" extensively discusses metabolic rhythms in eukaryotes but lacks information on the evolution of these rhythms from prokaryotes or their relevance to human microbiota.
5. Since the primary focus of this review is the evolution of the biological clock within the human microbiome and humans, it should be confined to discussions related to humans and their microbiota. Unnecessary explanations that do not directly contribute to this focus should be omitted.
6. Regarding Figure 1, it would enhance reader comprehension if the full names of entities in the figures (e.g., using "A" for archaea) are replaced with their complete names.
7. While the main focus of this article is the study of the biological clock's evolution from the human microbiota to the host, there is currently no figure illustrating this phenomenon. Figure 4, while useful, could be improved by adding a second panel that describes the evolution of the biological clock from the human microbiota to humans. This addition would enhance the review's clarity and reader engagement.
Comments on the Quality of English Language1. Carefully proofread the entire manuscript to rectify any typos or grammatical errors to enhance the overall quality of the document.
Author Response
Dear Reviewer
We appreciate your time spent reviewing our manuscript and for your pertinent comments. We greatly appreciate it. We have tried to address and discuss your comments as thoroughly as possible.
In response to the comment regarding anucleate cells, we would like to clarify that our review reports on circadian rhythms in anucleate cells, not limited to prokaryotic cells. This suggests the existence of time-keeping mechanisms that do not rely on genetic regulation in nucleate-free cells, which is why we used the term "anucleate". We refer here mainly to the discovery of circadian rhythms in erythrocytes.
Referring to the second paragraph, we have added key references in this section that relate to the discovery of the circadian clock in K. aerogenes and B. subtilis.
In reference to the third comment, our goal was to indicate that the discovery of a circadian clock in bacteria may have a beneficial impact on antibiotic therapy strategy. Understanding the activity of the circadian clock allows antibiotics to be administered at specific times of the day, potentially increasing drug efficacy. We have made therefore developments that highlight how knowledge of bacterial biological clocks can be one of the factors contributing to increased treatment efficacy.
The passage discussed in lines 1031-1138 (now 1058-1167) is intended to demonstrate the importance of metabolic regulation for the functioning of biological clocks. We are aware that most of the information in this paragraph applies to eukaryotic cells, with a much smaller portion relating to prokaryotic cells. This is primarily due to the lack of reliable literature on the evolution of metabolic rhythms in prokaryotic cells. Furthermore, many of the studies we cite represent discoveries made in the last few years. The goal of this review is also to inspire researchers to delve deeper and investigate this topic. Therefore, we hope that in the future, more studies will investigate the potential evolution of metabolic rhythms towards circadian clocks in prokaryotes.
With reference to the comment on the main theme of the review, we would like to correct that this work focuses primarily on the evolution of biological clocks, and the supporting narrative uses insights gained from studies of the microbiome. Research related to the discovery of circadian mechanisms in non-photosynthesizing bacteria, found in the human microbiome, form the basis of this review, as it has provided a comprehensive understanding of the topic at multiple levels.
We have made changes to the review to increase the clarity of the graphics and standardized them to reach a wider audience. Suggested changes have been incorporated into the graphics.
Regarding comment 7, we would like to emphasize that this work focuses on addressing the key questions related to the evolution of biological clocks, as outlined in the introduction. We acknowledge that this may not have been sufficiently highlighted in the introduction, so we have modified the sentence to clarify this.
Nonetheless, we have decided to improve this figure by creating a new one that better illustrates the topic of the evolution of biological clocks, now referred to as figure (x). Given that the text covers topics related to microbiology, the evolution of the human biological clock is outside the scope of this review. We have modified the graphic to make it more engaging for the reader while including more details related to the history of the evolution of biological clocks, with the most advanced versions observed in mammals and humans.
We carefully reviewed the entire manuscript and made corrections to improve the quality of the text.
Sincerely yours,
Adam Siebieszuk
Monika Sejbuk
Anna Witkowska
Reviewer 2 Report
Comments and Suggestions for Authors
The article "Studying the human microbiota: advances in understanding the fundamentals, origin, and evolution of biological timekeeping" explores the intriguing connection between the human microbiome and circadian rhythms, shedding light on the complex relationship between humans and microorganisms, and its implications for health. The discovery of circadian oscillations in the gut microbiota and non-photosynthetic gut bacteria has raised essential questions about the existence of functional biological clocks in microorganisms. While the article provides a commendable overview of the evolution and historic discoveries in molecular chronobiology, highlighting the mechanisms driving microbial biological clocks and their potential medical applications, several aspects warrant critical review.
The figures inserted in the article appear to be designed more like animations for a younger audience rather than adhering to the conventional scientific illustration style expected in academic publications. This departure from the typical scientific illustration format might raise concerns about the article's adherence to academic standards. In academic research, especially in reviews as the present one, figures serve a crucial role in visually representing data, concepts, or processes, and they are typically expected to be detailed, precise, and scientifically accurate. The use of cartoon-style figures might lead to questions about the professional quality and scientific rigor of the article. It is essential to maintain a balance between engaging and informative figures and maintaining a level of professionalism consistent with academic writing. Therefore, the inclusion of more traditionally formatted scientific figures might help to strengthen the overall scientific credibility of the article.
Furthermore, certain figures are really necessary but not provided, hindering the scientific clarity of the article. Figures are pivotal in conveying complex information and enhancing the understanding of readers. The absence of these figures may leave readers grappling with the concepts, thereby undermining the article's effectiveness. I would suggest that the authors add a comprehensive guide, together with a scientifically designed and explanatory figure for the present unanimously acknowledged mechanisms implied by the circadian entrainment process.
One more notable aspect is the excessive number of references (458!). While extensive referencing is generally a valuable attribute in academic writing, it's important to ensure that each reference is directly relevant to the content and adds substantial value. The article's over-reliance on references could be seen as overwhelming and could potentially be streamlined for a more concise and reader-friendly experience.
Overall, the article is well-written and touches upon significant aspects of microbial biological clocks and their potential applications in health. However, it may benefit from a more streamlined use of references, ensuring that they directly support the content, as well as the inclusion of the missing figures and the replacement of the existing ones in order to enhance scientific clarity. Addressing these issues would contribute to a more effective and academically rigorous presentation.
Comments on the Quality of English LanguageNo major concerns.
Author Response
Dear Reviewer
We appreciate your time spent reviewing our manuscript and for your pertinent comments. We greatly appreciate it. We have tried to address and discuss your comments as thoroughly as possible.
In response to the feedback on the graphics included in the manuscript, we have followed the suggestions and designed more readable illustrations. We removed unnecessary or overly detailed graphic elements (that were not relevant to the work) in the figures and made the images more informative, i.e. by adding the necessary textual annotations. In addition, following the feedback, we have decided to change the visual design and, in particular, the color scheme of virtually all graphics to better match scientific standards. We hope that the graphics are now much clearer and can reach a wider audience.
In order to better communicate the information contained in the text, we have made significant changes to some of the graphics, including separating them and creating more detailed versions, as well as developing completely new graphics. For example, in a newly developed graphic, we have depicted mechanisms involved in the complex activities of peroxiredoxins.
As far as the term ‘comprehensive guide’ is concerned, it is our understanding that Figure 2 fulfills the aforementioned criteria by illustrating the variety of possible mechanisms of biological clock. In addition, in order to provide a more detailed presentation of the evolutionary process and the factors influencing the evolution of biological clocks, we decided to split Figure 3 into two separate graphics, so that the newly created graphic (xx) covers a wider range of information, including highlighting the importance of the circadian entrainment process as a key driver of the evolution of timekeeping mechanisms.
We carefully reviewed the entire manuscript and managed to reduce the number of citations to 432, eliminating those with repeated information or citations where the literature may have been outdated. In addition, we corrected individual errors in the order of citation data (sorting).
The high number of citations is due to the comprehensive analysis of the topic and the broad approach to the issue at hand. This is a very comprehensive review, which is why addressing so many issues was necessary to accurately depict the entire landscape of the evolution of biological clocks.
Sincerely yours,
Adam Siebieszuk
Monika Sejbuk
Anna Witkowska
Round 2
Reviewer 1 Report
Comments and Suggestions for Authors
Please write the conclusion/summary in a separate section at the end
Reviewer 2 Report
Comments and Suggestions for Authors
Dear authors,
I definitely appreciate the efforts you took in order to perform the pointed observations, and I do consider that they have substantially improved the overall readability and comprehensibility of this review. It is a valuable work that will add value to this particular scientific field.